# ERBB3-dependent AKT and ERK pathways are essential for atrioventricular cushion development in mouse embryos

**Kyoungmi Kim** [ID] [1]*, **Daekee Lee** [2]*

1 Department of Physiology and Department of Biomedical Sciences, Korea University College of Medicine, Seoul, Republic of Korea, 2 Department of Life Science, Ewha Womans University, Seoul, Republic of Korea

* kim0912@korea.ac.kr (KK); daekee@ewha.ac.kr (DL)

## Abstract

ERBB family members and their ligands play an essential role in embryonic heart development and adult heart physiology. Among them, ERBB3 is a binding partner of ERBB2; the ERBB2/3 complex mediates downstream signaling for cell proliferation. ERBB3 has seven consensus binding sites to the p85 regulatory subunit of PI3K, which activates the downstream AKT pathway, leading to the proliferation of various cells. This study generated a human *ERBB3* knock-in mouse expressing a mutant ERBB3 whose seven YXXM p85 binding sites were replaced with YXXA. *Erbb3* knock-in embryos exhibited lethality between E12.5 to E13.5, and showed a decrease in mesenchymal cell numbers and density in AV cushions. We determined that the proliferation of mesenchymal cells in the atrioventricular (AV) cushion in *Erbb3* knock-in mutant embryos was temporarily reduced due to the decrease of AKT and ERK1/2 phosphorylation. Overall, our results suggest that AKT/ERK activation by the ERBB3-dependent PI3K signaling is crucial for AV cushion morphogenesis during embryonic heart development.

## Introduction

During mouse cardiogenesis, the linear heart tube from the cardiogenic mesoderm forms the outflow tract (OFT) and atrioventricular (AV) cushion after the looping, and endocardial cells initiate the endocardial to mesenchymal transformation (EndMT). After forming the endocardial cushion by the EndMT, mesenchymal cells in the cushion start to proliferate rapidly for valve formation. Furthermore, the superior and inferior AV cushions fuse to form a septum, dividing the ventricular inlet into right and left chambers. Meanwhile, the OFT cushion region gives rise to the arterial pole, and the myocardium of the ventricular becomes trabeculated.

Gene targeting experiments in mice reveal that ERBB family members and their ligands play an essential role in embryonic heart development and adult heart physiology [1–4]. The ERBB family includes ERBB1 (EGFR), ERBB2, ERBB3, and ERBB4, single transmembrane receptor tyrosine kinases that homodimerize or heterodimerize with other family members upon ligand binding to the extracellular domain. Subsequently, the phosphorylation of the cytoplasmic tail recruits adaptors to activate various downstream context-specific signaling

**Data Availability Statement:** All relevant data, including original uncropped and unadjusted images underlying all blot or gel results, is within the paper and Supplementary data files.

**Funding:** This work was supported by the National Research Foundation of Korea (2011-0009719,

2020M3A9D5A01082439, and 2019R1A2C2087198).

**Competing interests:** The authors have declared that no competing interests exist.

pathways [5]. In contrast to the other members, ERBB3 has seven consensus binding sites to p85, the regulatory subunit of PI3K, which activates the downstream AKT pathway [6–8] and leads to the proliferation of a wide variety of cells. Despite its low kinase activity, ERBB3 is a preferential binding partner of ERBB2; the ERBB2/3 complex mediates downstream signals for cell proliferation [9, 10].

In this study, we generated a human *ERBB3* knock-in mouse expressing a mutant ERBB3 whose seven YXXM p85 binding sites were replaced with YXXA. This human ERBB3 knock-in mouse exhibits a deficiency of seven consensus binding sites for p85 to uncouple the ERBB3-dependent PI3K pathway. Furthermore, we analyzed the ERBB3-dependent PI3K pathway for AV cushion morphogenesis during embryonic heart development in knock-in mouse.

## Materials and methods

### Expression vectors and site-directed mutagenesis

Full-length human *ERBB2* and *ERBB3* cDNAs were subcloned into *pcDNA3.1-* vector (Thermo Fisher Scientific) to obtain *pERBB2* and *pERBB3* expression vectors, respectively. Seven YXXM PI3K p85 consensus binding sites in human *ERBB3* cDNA underwent site-directed mutagenesis using the QuickChange Multi Site-Directed mutagenesis kit as described by the manufacturer (Agilent Technologies). The *pERBB3*$^{7A}$ vector was generated by changing methionine at positions 944, 1057, 1200, 1225, 1263, 1279, and 1292 to alanine [5], whereas the *pERBB3*$^{7F}$ vector was generated by changing tyrosine at positions 941, 1054, 1197, 1222, 1260, 1276, and 1289 to phenylalanine [11]. DNA sequencing revealed that there was no change in the nucleotide sequences except at mutagenized sites. The flag was inserted to the c-terminal ERBB3 or ERBB3$^{7A}$ expression vectors to obtain *pERBB3-Flag* or *pERBB3*$^{7A}$*-Flag* vectors.

### Generation and establishment of Erbb3 mutant mice

The *Erbb3* null allele without exon 2 (*Erbb3*$^{-}$) was generated by pronuclear microinjection with 1 μg/mL of the circular *pCreEGFP* plasmid (expression of Cre recombinase fused with EGFP) into a heterozygous 1-cell embryo containing the conditional *Erbb3* allele (*Erbb3*$^{flox}$) [12]. To construct the *ERBB3*$^{7A-Neo}$ knock-in vector, a 9-kb *Hind*III fragment of *Erbb3* genomic DNA in 129SV BAC clone spanning exon 1 to exon 3 was subcloned into *pBluescript* II vector. The *Sac*II site in exon 1 and *Bsr*G1 in exon 2 was used for subcloning of the 5' homology arm, and the 3' homology arm was derived from the *Xho*I fragment of a conditional *Erbb3* targeting vector described previously [12]. To add the *polyA* signal, the 0.7-kb 3' untranslated region (UTR) of mouse *Erbb3* was PCR cloned into the *Bsr*G1 site right after the 5' homology arm. Lastly, *Bsr*G1 and *Not*I fragments of *pERBB3*$^{7A}$ vector were subcloned between the 5' homology arm and 3'UTR. So, the final fused *Erbb3* gene consists of a mouse gene with the *Bsr*G1 site in exon 2 and a human gene (from *Bsr*G1 site to stop codon of *ERBB3*$^{7A}$ cDNA) with mouse *polyA* signal. Targeting in 129/SvEv embryonic stem (ES) cells and generation of a chimeric mouse were described previously [12]. Germline transmission of the *Erbb3*$^{7A-Neo}$ allele was verified with PCR. To remove the *neomycin transphosphorylase* (Neo) expression cassette, one-cell embryos derived from C57BL/6J (B6) females mated with *Erbb3*$^{+/7A-Neo}$ males were treated with 3 μM of purified HTNCre protein for 30 min as described previously [13]. Complete excision of the Neo selection marker was identified from the founder mice by PCR genotyping with primers p489, 5'-ACTGAACCCCACCTACATTGTC-3' (sense, S) and p150, 5'-AAGCCTTCTCTATGGAAAGTG-3' (antisense, AS). Accurate *Bsr*GI junction between mouse and human *ERBB3* cDNA was verified using RT-PCR product from the *Erbb3*$^{7A/7A}$ embryo. The genotype of each mouse was determined by PCR using DNA extracts

from the yolk sac or toe clip with the following primers: p149, 5′-TCCAGCGTGGAAAAGTTC AC-3′ (S); p150, 5′-AAGCCTTCTCTATGGAAAGTG-3′ (AS); p680, 5′-CCCTGACAGAAT CTCGGTGA-3′ (AS). *Erbb3⁺ᐟ⁻* or *Erbb3⁺ᐟ⁷ᴬ* mice were backcrossed more than ten generations with the B6 background before intercross. Mice were housed in a specific-pathogen-free facility, and all experiments with mice were approved by the Institutional Animal Care and Use Committee at Ewha Womans University.

## RNA preparation and quantitative real-time PCR (qRT-PCR)

Total RNA from an E10.5 embryo was prepared, and qRT-PCR and quantification of mRNA were performed as described previously [14]. The sequence of qRT-PCR primers used were: *Egfr*, 5′-GCAAAGTGCCTATCAAGTGGA-3′ (S), 5′-CCAGCACTTGACCATGATC-3′ (AS); *Erbb2*, 5′-CAGATTGCCAAGGGGATGA-3′ (S), 5′-TGCCCCCATCTGCATGGTA-3′ (AS); *Erbb3*, 5′-CCACAGCTGCTGCTCAACT-3′ (S), 5′-AATTGGAGTCTTGGCCTCAC-3′ (AS); *Erbb4*, 5′-ACTGCTGCCATCGAGAATG-3′ (S), 5′-CCAGTTGAAAGGTGGTTGG-3′ (AS); *18S rRNA*, 5′-TCAACTTTCGATGGTAGTCGCC-3′ (S), 5′-GGCCTCGAAAGAGTCCTGT ATTGT-3′ (AS). RT-PCR was performed as described previously [12] with primers p268, 5′-TTCCGAGATGGGCAACTCTC-3′ (S) and p665, 5′-ACTTCCCATCGTAGACCTGG-3′ (AS). Primer p268 is specific for the mouse *Erbb3* sequence, whereas p665 is conserved in both human and mouse.

## Histology and immunohistochemistry of embryonic samples

Embryos were fixed in 10% NBF at 4˚C overnight and embedded in paraffin. Rehydrated paraffin sections were stained with hematoxylin and eosin Y (H&E) and mounted with Permount solution (Thermo Fisher Scientific). The embryo images were visualized with a light microscope (Eclipse 80i EPI, Nikon or Axiovert 200, Zeiss). For cryosection, the embryos were fixed in 4% PFA in 1× PBS at 4˚C overnight, equilibrated in 30% sucrose in 1× PBS at 4˚C, and embedded in OCT. Paraffin sections or cryosections were used for immunohistochemistry. For immunofluorescence staining, the embryo sections were blocked with 5% normal goat serum in TBST (10 mM Tris-HCl, pH 7.4, 150 mM NaCl, 0.1% Tween 20), then incubated overnight at 4˚C with primary antibodies. For detecting a target with a mouse primary antibody, M.O.M kit (Vector Labs) was used according to the manufacturer's protocol. The embryo sections were further treated with secondary antibodies conjugated with Alexa Fluor 488 or 568 (Life Technologies) for 1 h or treated with secondary antibody conjugated with biotin for 30 min, followed by treatment with fluorescein avidin DCS (Vector Labs) for 5 min to intensify the signal. The sections were counterstained with DAPI and mounted with Vectashield medium (Vector Labs). The images were visualized with an LSM 510 Meta confocal microscope (Zeiss) with the same exposure intensity value. The immunofluorescence images were quantified with Nikon NIS-Elements BR 3.2 imaging software. In a sample, each cell's area and background were assigned, and their intensities were measured and calculated as average intensity, excluding background intensity.

## BrdU staining

Embryos were labeled with BrdU (Sigma-Aldrich) for 2 h after intraperitoneal injection with 10 mM BrdU in PBS, pH 7.4, at 0.1 mL/10 g body weight into pregnant females and embedded as described in Methods. Paraffin sections were immunostained, as described previously [12].

## Collagen gel analysis of EndMT in the embryo heart

The AVC region of the heart at the E9.5 embryo stage was dissected out and opened with fine iris scissors under a dissecting microscope. The endothelial surface was placed toward collagen gel containing 1 mg/mL collagen I from rat tail (Thermo Fisher Scientific) in Medium 199. The explant was allowed to attach at 37˚C in a humidified atmosphere of 5% $CO_2$ overnight and was cultured in 0.1 mL of Medium 199 supplemented with 1% FBS, 0.01% Insulin-Transferrin-Selenium (Thermo Fisher Scientific), and penicillin/streptomycin at 37˚C in a humidified atmosphere of 5% $CO_2$ for 48 h [15]. Transformed cells showing stellate and spindle-shapes were counted under an inverted light microscope (Axiovert 40C,) as described previously [15–17]. PI3K inhibitor LY294002 (Merck KGaA) and MEK inhibitor PD0325901 (Selleckchem) were dissolved in DMSO, and an equal amount of stock solution was added to the culture media (final DMSO concentration, 0.2%).

## CHO cell culture and establishment of ERBB2 stable cell line

CHO cells were maintained in DMEM medium supplemented with 10% FBS at 37˚C in a humidified atmosphere of 5% $CO_2$. Stable colonies that expressed ERBB2 were established after transfection with the *pERBB2* vector using LT1 reagent (Mirus), followed by selecting 250 µg/mL of Geneticin (Thermo Fisher Scientific) for 2 weeks. The ERBB2 expression level was examined with western blotting, and a representative stable cell line (CHO-ERBB2) was maintained.

## Cell transfection

CHO-ERBB2 was seeded on 6-well plates in a density of 0.75 x $10^5$ cells and was transiently transfected with *pERBB3*, *pERBB3$^{7A}$,* and *pERBB3$^{7F}$* vectors using LT1 17–24 h after seeding to express *ERBB3*, *ERBB3$^{7A}$,* and *ERBB3$^{7F}$*, respectively. After 30 h of transfection, transfected cells were starved for 18 h in DMEM containing 0.1% FBS then treated with 50 ng/mL of rhNRG1-β1 (R&D Systems) for 5 min 37˚C. Cells were immediately washed with cold PBS and stored at -80˚C until preparation of protein extract.

## Western blot and immunoprecipitation

The protein extract of a whole embryo at E10.5 was prepared by homogenizing the embryo with a glass-Teflon homogenizer in tissue lysis buffer containing proteinase and phosphatase inhibitors. The protein extraction of the heart endocardial cushion and the trabecular region was performed by first dissecting the fetal heart at E11.5 with fine iris scissors and then freezing the samples in dry ice. A piece of tissue was homogenized with a microtube pellet pestle (Sigma-Aldrich) to prepare the cell extracts [12]. Centrifuging the cell lysate to separate the supernatant from the pellet, protein quantification, SDS-PAGE, western blotting, and quantification of blots were performed [12].

The detailed procedure for immunoprecipitation was described previously [18].

## Antibodies

For immunostaining, the following antibodies were used: anti-p-AKT (S473, #4060) and anti-p-ERK1/2 antibodies (#4370) from Cell Signaling Technology; anti-BrdU antibody from Abcam (ab6326); anti-SNAIL antibody (sc28199) from Santa Cruz; and anti-CD144 (VE-cadherin) antibody from BD Pharmingen (550548). For western blotting or IP (Immunoprecipiation), the following antibodies were used: anti-ACTB (AC-15) and anti-Flag antibody (F3165) from Sigma-Aldrich; anti-AKT (#9272), anti-AKT(S473, #4060), anti-p-AKT(T308, #2965),

anti-EGFR antibody (#2232), anti-p-ERBB3(Y1289, #4791), anti-ERK1/2 (#9102), anti-p-ERK1/2 (#9101) antibodies from Cell Signaling Technology; anti-ERBB2 (sc284-G), anti-ERBB3 (sc285-R) and anti-ERBB4 antibodies (sc283-R) from Santa Cruz Biotechnology; anti-GAPDH antibody from Abfrontier (LF-PA0018); and anti-p-Tyr antibodies from BD Biosciences (610024).

### Statistical analysis

Experimental groups were compared with a two-tailed Student's t-test or one-way ANOVA with Newman-Keuls multiple comparison test. Data represent mean ± SEM. $P < 0.05$ was considered statistically significant. Each letter (a, b, c or d) in the graph indicates a significant difference between experimental groups. Two letters that are the same (e.g., a and a) indicate a nonsignificant difference (ns), whereas different letters (e.g., a and b) indicate a significant difference. Two letters together (e.g., ab) indicates non-significant differences (ns) with either a or b, but a significant difference compared with c or d.

## Results

### $Erbb3^{7A/7A}$ mutant embryos show embryonic lethality

All seven YXXM consensus binding motifs were changed to YXXA in the human $ERBB3$ cDNA to uncouple the PI3K downstream pathway from ERBB3. The mutant human $ERBB3$ cDNA was used to construct the $Erbb3^{7A}$ knock-in mouse on the C57BL/6 background; the knock-in of the mutant $ERBB3$ was verified by measuring its mRNA level (Fig 1A–1E). Overall, the $Erbb3^{7A}$ allele was found to express the human ERBB3 protein, whose N-terminus was substituted with the first 48 amino acids of the mouse ERBB3 protein. qRT-PCR revealed that the human $ERBB3$ mRNA level in the $Erbb3^{7A/7A}$ embryos was comparable to the $Erbb3$ mRNA level in the wild-type embryos (Fig 1F). ERBB3 protein was not detected in $Erbb3^{-/-}$ embryo. However, the ERBB3 protein level in the $Erbb3^{7A/7A}$ embryos, similar to $Erbb3^{+/-}$, was only one-third of that in the wild-type embryos (Fig 1G). The protein levels of all other members were similar regardless of genotype. The transient transfection of Flag-tagged $ERBB3$ or Flag-tagged $ERBB3^{7A}$ revealed that the reduced level of ERBB3 protein in the $Erbb3^{7A/7A}$ embryo was not due to a possible epitope change in ERBB3$^{7A}$ protein (Fig 1H).

Next, we analyzed the viability of the $Erbb3^{7A/7A}$ embryos because $Erbb3^{7A/7A}$ pups were not found from the intercrosses of $Erbb3^{+/7A}$ mice (Fig 2A). The Mendelian ratio that 25% of the offspring were $Erbb3^{7A/7A}$ mutant embryos was maintained until E11.5. Then, the proportion of $Erbb3^{7A/7A}$ embryos decreased to 0% at E14.5. Similarly, no $Erbb3^{-/-}$ embryos on the C57BL/6 background survived to E14.5; these data were inconsistent with a previous finding that 21% of the $Erbb3^{-/-}$ embryos survived to term on a heterogeneous background [19]. These results suggest that the $Erbb3^{7A}$ allele and the $Erbb3^{-}$ allele result in the same level of lethality. Microscopic and histological examination results of E12.5 embryos showed that both $Erbb3^{7A/7A}$ and $Erbb3^{-/-}$ embryos have a small right ventricle (Fig 2B and 2C). Additionally, in the E13.5 $Erbb3^{-/-}$ and $Erbb3^{7A/7A}$ embryos, hemorrhage was observed at the site of contact between the left superior intercostal vein and left superior vena cava; the site was filled with blood (Fig 2D and 2E).

### Reduced EndMT in $Erbb3$ mutant embryos is associated with aberrant SNAIL and VE-cadherin protein levels

$Erbb3^{-/-}$ embryos lacked AV cushion mesenchymal cells at E9.5-E10, and AVC-explant culture demonstrated that the $Erbb3^{-/-}$ embryos had a defective endothelial to mesenchymal transition

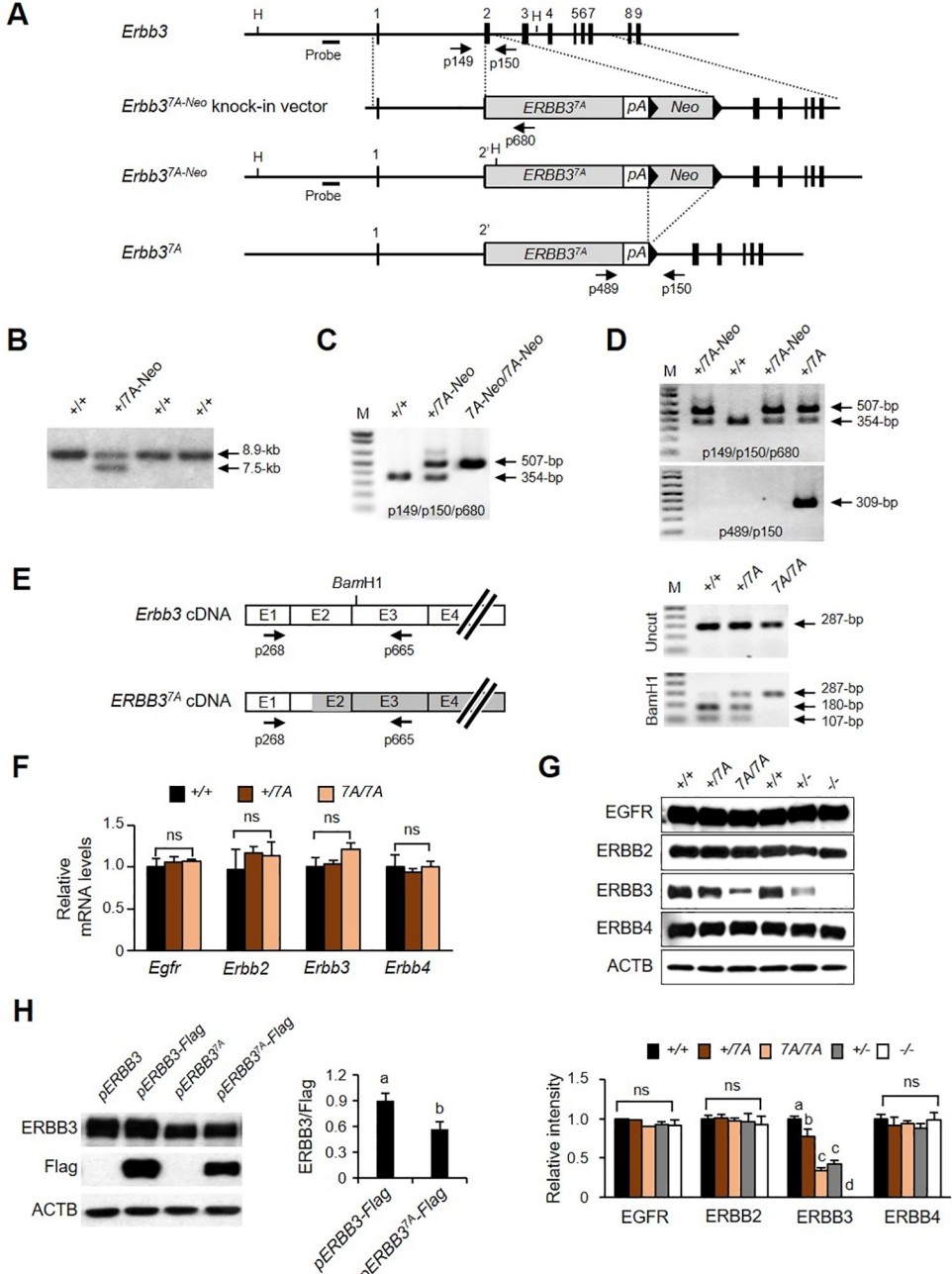

**Fig 1. Targeting strategy for *Erbb3* gene and characterization of *Erbb3* expression.** (A) *Erbb3* wild-type allele, targeting vector, and targeted *Erbb3^{7A-Neo}* allele are shown. The numbers 1–9 indicate exons. H indicates the *Hind*III site. *ERBB3^{7A}*, *pA*, and *Neo* in knock-in vectors represent human *ERBB3^{7A}* cDNA, the polyA sequence including the 3'UTR region of mouse *Erbb3* gene and neomycin transphosphorylase expression cassette, respectively. Arrowheads indicate *lox*P sites. 2' denotes the partial exon 2 of mouse gene. Arrows indicate PCR primers for genotyping. (B) Genomic DNA from ES cell clones was digested with HindIII and hybridized with a radiolabeled probe as shown in S1A Fig in S1 File, resulting in an 8.9-kb fragment from wild-type allele 7.5-kb fragment from the correctly targeted allele, respectively. (C) PCR genotyping using three primers (p149, p150, and p680 as shown in S1A Fig in S1 File) discerned the genotype of E10.5 embryos obtained from *Erbb3^{+/7A-Neo}* intercrosses, and primers p149 and p150 resulted in a 354-bp PCR product for wild-type *Erbb3* allele. In contrast, primers p149 and p680 resulted in a 507-bp PCR product for the *Erbb3^{7A-Neo}* allele. M, 1 kb Plus ladder. (D) PCR genotyping with indicated primers is shown using DNA from pups treated with HTNCre at the one-cell stage. Primers p489 and p150 resulted in a 309-bp PCR product specific for the *Erbb3^{7A}* allele. (E) Schematic drawing of cDNA from *Erbb3* and *Erbb3^{7A}* allele and analysis of RT-PCR product. E1 to E4 indicates exon 1 to exon 4. Open-box and gray box derive from mouse *Erbb3* and human

*ERBB3* cDNA, respectively. RT-PCR analysis, followed by *Bam*HI digestion, confirmed human ERBB3 mRNA expression from the *Erbb3^{7A}* knock-in allele. Representative agarose gel electrophoresis of RT-PCR product from E10.5 embryos is shown on the right. *Bam*HI digestion of a 287-bp RT-PCR product (primers with p268 and p665) resulted in 180-bp and 107-bp fragments in wild-type and heterozygous *Erbb3^{7A}* mutant embryos, respectively, whereas no fragments were shown in the *Erbb3^{7A/7A}* embryo. (F) Relative mRNA levels of *Egfr* family members were analyzed by qRT-PCR. (G) Protein levels of EGFR family members in a whole E10.5 embryo were analyzed by western blotting with indicated antibodies depending on genotypes (n = 5–6). (H) Western blotting was performed with cell lysates using the indicated antibodies. The transient expression of ERBB3 in CHO cells was achieved by transfecting cells with the indicated vectors. ACTB (Beta-Actin) was used as the loading control. Different letters on the bar graphs represent statistically significant differences, $p < 0.05$; ns, non-significant.

(EndMT) [10, 17, 20]. We first identified ERBB3 expression in the hearts of E10.5 stage embryos. ERBB3 was expressed in endothelial cells and mesenchymal cells of the AV cushion (S1 Fig in S1 File). Also, we performed histological analysis of the AV cushion in *Erbb3^{7A/7A}* embryos because *Erbb3^{7A/7A}* embryos have heart defects similar to *Erbb3^{-/-}* embryos. In E10.5 embryos of *Erbb3^{7A/7A}* and *Erbb3^{-/-}*, the number of mesenchymal cells in the AV cushion was reduced compared to the wild type (Fig 3A and 3B). The number of mesenchymal cells continued to increase in both types of mutant embryos, but at E11.5 their numbers were still lower than in wild-type embryos (Fig 3A and 3B). The number of endocardial cells per area showed no difference regardless of genotype (Fig 3C).

In wild-type AVC-explants, the outgrowths of endocardial cells appeared widely spread around the collagen gel surface, but the both *Erbb3^{7A/7A}* and *Erbb3^{-/-}* mesenchymal cells did not (Fig 3D). The AVC-explants in both mutant embryos had a reduced number of mesenchymal cells by 40%–60% as compared to wild-type embryos (Fig 3D). The VE-cadherin, a key player of EndMT [21], and SNAIL, one of the transcriptional repressors of VE-cadherin [21–23], were analyzed using immunofluorescence staining in AVC of E9.5 embryos to determine the cause of the reduced EndMT (Fig 3E). Overall, the VE-cadherin protein was distributed more broadly in both types of *ErbB3* mutant embryos than wild-type embryos (Fig 3E).

In contrast to VE-cadherin's weak immunostaining between the endothelial and delaminated cells in wild-type embryos, we more frequently detected intense VE-cadherin immunostaining between the cell types in mutant embryos. These results suggest the possibility that attenuation of EndMT is related to high VE-cadherin levels (Fig 3E). Moreover, round and oval nuclei were found in the AVC of the mutant embryos, whereas thin and long nuclei were found in the wild-type endocardial cells. The SNAIL protein was mainly found in the endocardial cells with thin and long-shaped nuclei and less frequently in the endocardial cells with round and oval nuclei (Fig 3E).

## Proliferation of mesenchymal cells is transiently attenuated in the AV cushion of *Erbb3* mutant embryos

After EndMT, the transformed mesenchymal cells undergo extensive proliferation in the AV cushion before AVC morphogenesis [24, 25]. We performed BrdU immunostaining to investigate cell proliferation in the heart (Fig 4A). At E10.5, approximately 43% of mesenchymal cells in the wild-type embryos were BrdU-positive. However, the BrdU-positive cells were reduced to approximately 31% in mutant embryos (Fig 4B). BrdU-positive mesenchymal cells were temporarily weakened at E10.5 of the two mutant embryos, but there was no difference at E11.5 stage. In contrast, the number of BrdU-positive endocardial cells in the wild-type and mutant embryos at E10.5 were similar. However, their numbers decreased in both mutant embryos at E11.5 (Fig 4C).

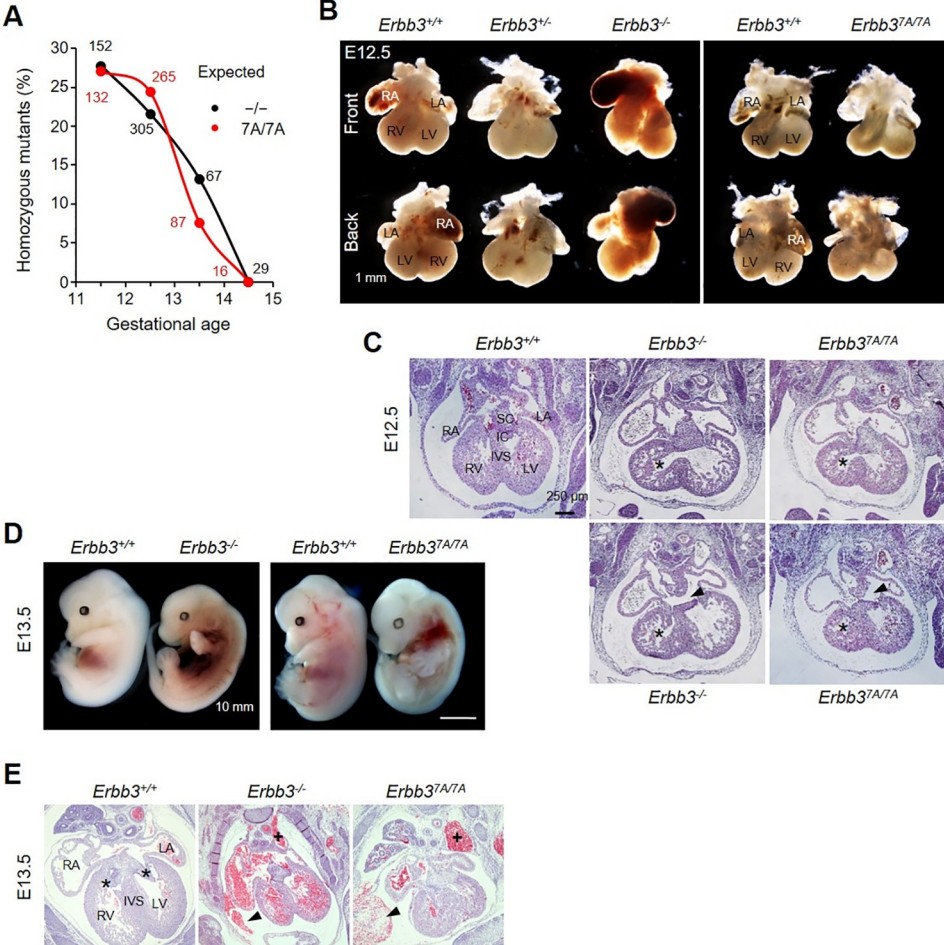

**Fig 2. Similar heart defects in *Erbb3^7A/7A* and *Erbb3^-/-* embryos.** (A) Genotype was determined by PCR analysis of embryos. The viability of *Erbb3* mutant embryos from heterozygous intercrosses was determined through the heartbeat of embryos. The Mendelian ratio of homozygous mutant embryos is shown as a dotted line. The number denotes the total number of embryos genotyped. (B) Shown are E12.5 hearts of wild-type, *Erbb3^+/-*, *Erbb3^-/-* and *Erbb3^7A/7A* embryos observed under a dissecting microscope. Arrows indicate a small right ventricle of the mutant embryos. (C) Representative H&E staining of the E12.5 heart sections. Asterisks indicate the right ventricle, arrows indicate the opening of the interventricular septum, and arrowheads indicate the non-fused superior and inferior endocardial cushion of mutant embryos, respectively. (D) Shown are E13.5 wild-type, *Erbb3^-/-* and *Erbb3^7A/7A* embryos observed under a dissecting microscope. Arrows indicate bleeding in the mutant embryos. (E) Representative H&E staining of the E13.5 heart sections. Asterisks indicate the right and left AV valve, respectively. Arrowheads indicate bleeding in the pericardial cavity, and pluses (+) mark the site of communication between the left superior intercostal vein and left superior vena cava filled with blood in the mutant embryos, respectively. IC, inferior endocardial cushion; IVS, interventricular septum; LA, left atrium; LV, left ventricle; RA, right atrium; RV, right ventricle; SC, superior endocardial cushion.

## AKT and ERK1/2 phosphorylation is attenuated in the AV cushion of *Erbb3* mutant embryos

Western blot analysis was performed with protein extracts from individual AV cushions at E11.5 to identify the molecular mechanism of Nrg1-ERBB3 signaling for embryonic heart development. AKT and ERK pathways are two major downstream pathways of ERBB3 signaling [26, 27]. The relative levels of phosphorylated AKT (p-AKT(S473)) and ERK1/2(p-ERK1/2) were lower in both mutant embryos than in the wild-type, although their total protein levels did

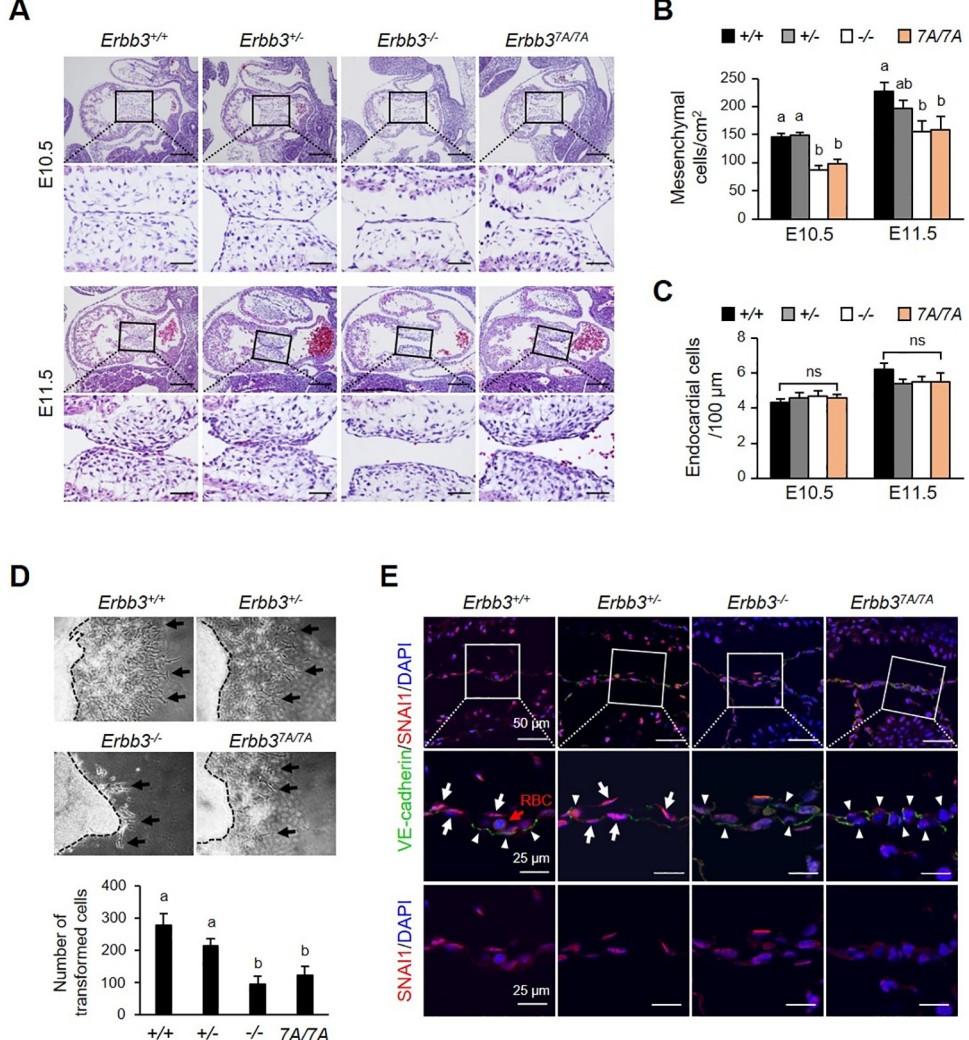

**Fig 3. The attenuation of EndMT in the heart of *Erbb3* mutant embryos.** (A) The atrioventricular (AV) cushion in the heart at E10.5 and E11.5 was examined with H&E staining. Scale bars indicate 200 μm and 50 μm, respectively. (B) The number of mesenchymal cells in the AV cushion in the samples was quantified (n = 5–8). (C) The endocardial cells in the AV cushion in the samples were quantified (n = 4–7). (D) EndMT analysis via the collagen gel method of the AVC-explants at E9.5. Arrows indicate the transformed mesenchymal cells. The bottom graph displays the quantification of the transformed cells. (E) Immunofluorescence staining of VE-cadherin and SNAIL in the AV cushion at E9.5 heart. Arrows indicate endocardial cells showing the SNAIL protein mainly located in the nucleus, whereas arrowheads indicate SNAIL proteins in the cytoplasm. Nuclei were counterstained with DAPI. RBC, red blood cell. The bar graphs represent the mean ± SEM, and different letters on the bar represent statistically significant differences, $p < 0.05$; ns, non-significant. Each letter of a or b in the bar graph means a significant difference ($p < 0.05$) for each experimental group, and ab means a non-significant difference for experimental groups.

not differ (Fig 5A and S2 Fig in S1 File). Next, we performed immunostaining with phospho-specific antibodies in the AV cushion at E10.5 and E11.5. AKT and ERK1/2 were mainly phosphorylated in the endocardial and mesenchymal cells of the AV cushion. However, the phosphorylation levels were decreased in both mutant embryos (Fig 5B and 5C; Top). The quantification of phosphorylation intensity in the endocardial or mesenchymal cells of the AV cushion revealed that p-AKT or p-ERK1/2 was significantly reduced in both mutant embryos compared to the wild-type (Fig 5B and 5C; Bottom).

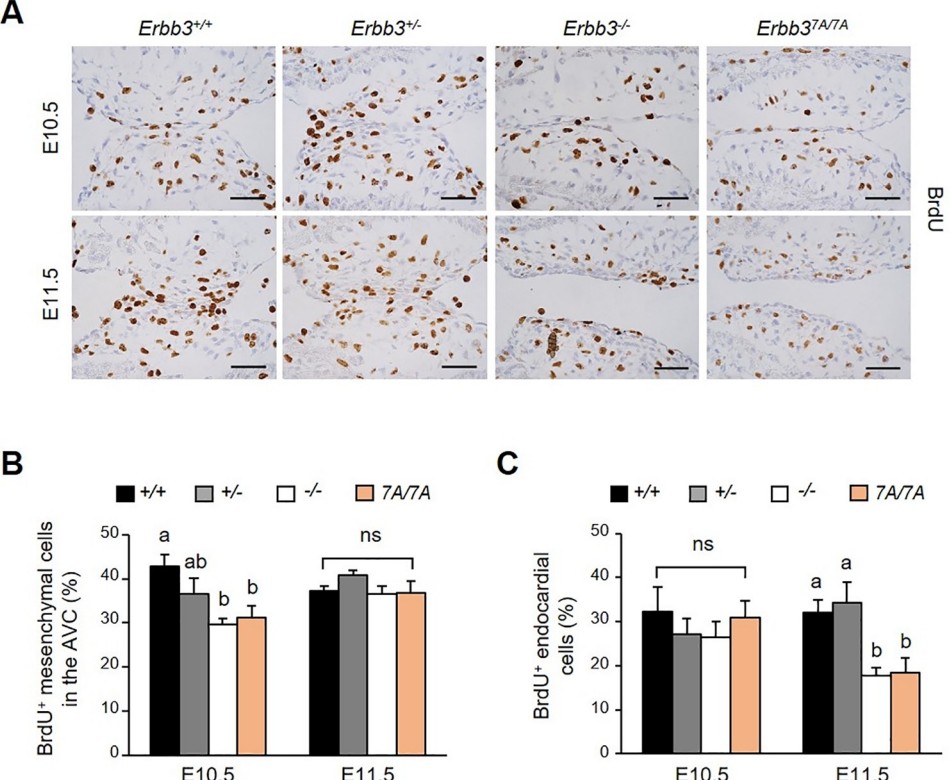

**Fig 4. Reduced proliferation of endocardial and mesenchymal cells of AV cushion in *Erbb3* mutant embryos.** (A) AV cushion was immunostained with anti-BrdU antibody. Scale bar indicates 50 μm. (B) The number of BrdU-positive mesenchymal cells in the AV cushion was counted and quantified (n = 5–7). (C) The number of BrdU-positive endocardial cells in the AV cushion was counted and quantified (n = 5–7). Different letters on the bar graphs represent statistically significant differences, $p < 0.05$; ns, non-significant.

PI3K and ERK pathways inhibitors blocked the EndMT in AVC-explant culture [28, 29]. We tested co-treatment of the AVC explant with PI3K and MEK inhibitors to determine whether the reduced expression of the PI3K and MEK pathways by ERBB37A mutation affected early EndMT. The co-treatment of the AVC explant with PI3K and MEK inhibitors achieved higher EndMT inhibition within the AV cushion than did treatment with one inhibitor (S3A, S3B Fig in S1 File). These results indicate that the embryonic heart defect in Erbb3 mutant embryos was due to the reduction in early EndMT and attenuation of both AKT and ERK phosphorylation.

## AKT and ERK pathways are attenuated by ERBB3<sup>7A</sup>, but not by ERBB3<sup>7F</sup> in CHO cells

Lahlou H et al. reported the generation of a *Erbb3<sup>Δ85</sup>* knock-in mouse, in which the seven YXXM PI3K p85 consensus binding sites of human ERBB3 were replaced with FXXM. In contrast to *Erbb3<sup>7A</sup>* mice, some mice carrying the homozygous *Erbb3<sup>Δ85</sup>* allele survived into adulthood [11]. This indicates that their ERBB3 mutant proteins may induce differential downstream pathways. However, both proteins have mutations in the p85 binding sites; therefore, we set out to identify their different effects on downstream signaling. We generated a *pERBB3<sup>7F</sup>* (YXXM to FXXM) vector by replacing the wild-type amino acid sequences of human ERBB3 with the corresponding mutant sequence in the *Erbb3<sup>Δ85</sup>* allele. We transfected the mutant construct into an ERBB2-expressing CHO cell line (CHO-ERBB2) [30].

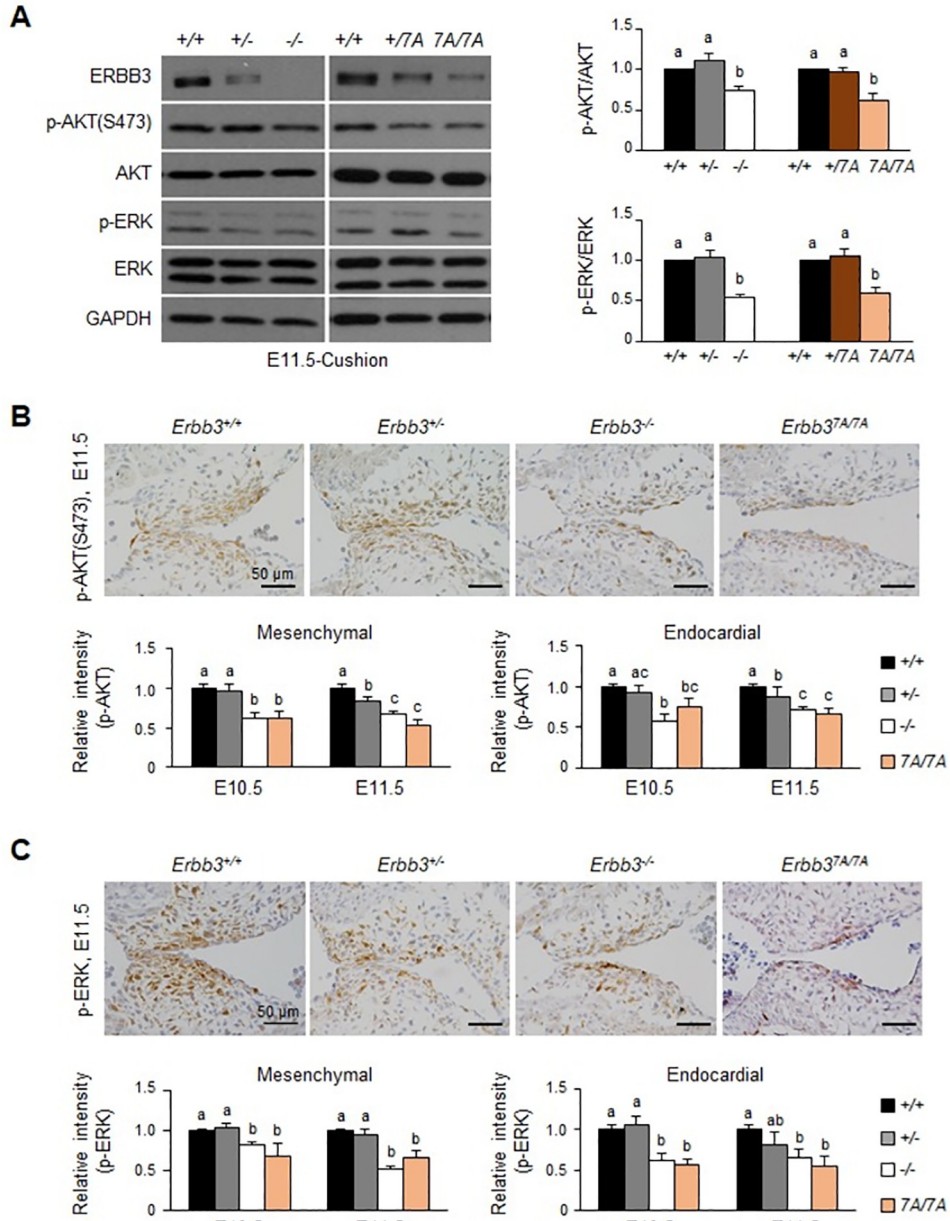

**Fig 5. Reduction of AKT and ERK phosphorylation in the heart of *Erbb3* mutant embryos.** (A) (Left) The relative levels of AKT and ERK phosphorylation in the endocardial cushion as determined by western blotting. GAPDH was used as the loading control. (Right) The band intensities of p-AKT(S473) and p-ERK1/2 were quantified and compared to the wild-type (n = 6–10). (B) p-AKT or (C) p-ERK1/2 in the AV cushion at E11.5 was examined by immunostaining with the indicated antibodies. The immunostained images in mesenchymal and endocardial cells were quantified using Nikon NIS-Elements BR3.2 imaging software (n = 4–9) to analyze the relative level of AKT or ERK1/2 phosphorylation. Different letters on the bars represent a statistically significant difference between groups, $p < 0.05$. Each letter of a or b or c on the bar graph means a significant difference ($p < 0.05$) for each experimental group, and ab means a non-significant difference for experimental groups.

Western blotting using the anti-phosphorylated ERBB3(p-ERBB3) antibody detects the phosphorylation of ERRB3 at Tyr 1289, located in the far end of the p85-binding site of the C-terminus. The level of NRG1β-induced ERBB3 phosphorylation and subsequent AKT and

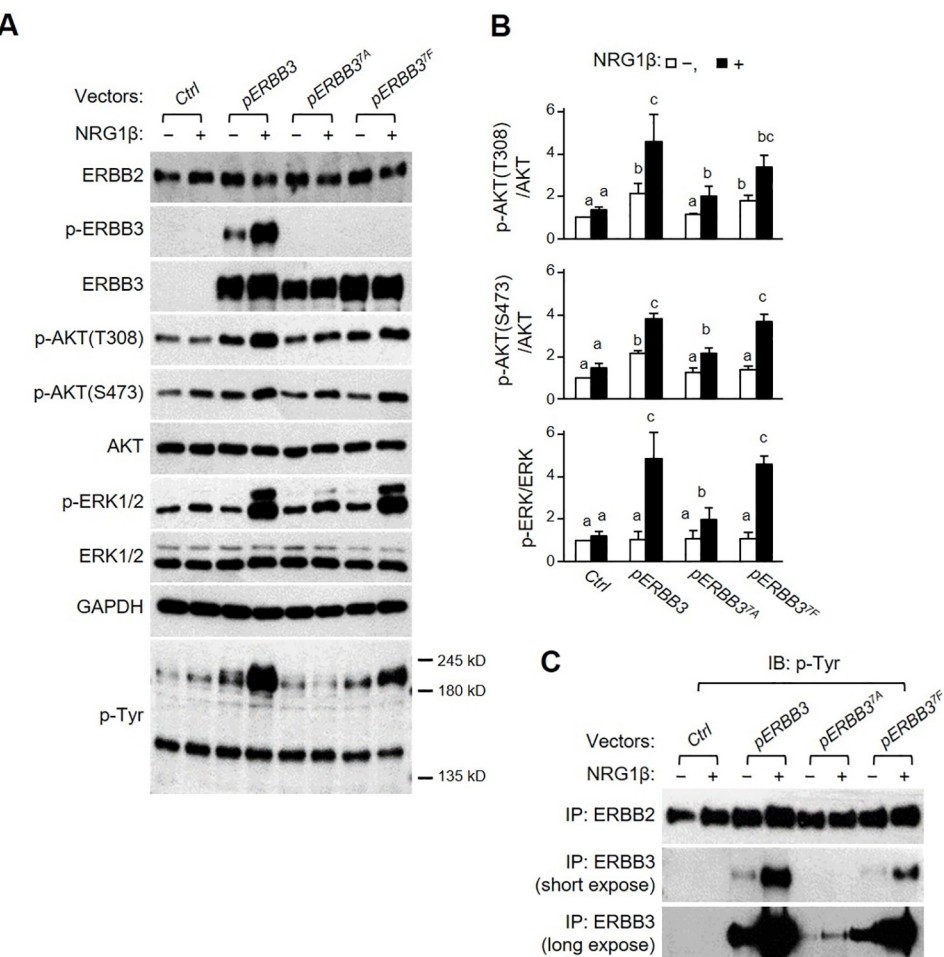

**Fig 6. Differential induction of downstream signaling by ERBB3 mutant proteins in CHO cells.** (A) CHO cells stably expressing ERBB2 (CHO-*ERBB2*) were transiently transfected with *pcDNA3.1 (Ctrl)*, *pERBB3*, *pERBB3^{7A,}* or *pERBB3^{7F}* vectors. Transfected cells were treated with NRG1β (+) or vehicle (-), and the whole-cell lysates were subjected to western blotting using the indicated antibodies. (B) The relative intensity of the bands in western blots was compared to the *Ctrl* vector without NRG1β treatment. Each different letter on the bars represents statistically significant differences between groups, $p < 0.05$, and bc means a non-significant difference for experimental groups. (C) The cell lysates were immunoprecipitated (IP) with the anti-ERBB2 or anti-ERBB3 antibodies, and IP samples were further analyzed by western blotting (IB) using an anti-p-Tyr antibody.

ERK1/2 phosphorylation were dramatically increased in the cells transfected with the *pERBB3* vector (Fig 6A and 6B). In contrast, the level of NRG1β-induced p-AKT and p-ERK1/2 was profoundly attenuated in *pERBB3^{7A}*-transfected cells despite a slight increase in phosphorylation by NRG1β treatment. Despite the absence of phosphorylated ERBB3 (p-ERBB3) in FXXM mutation human ERBB3 (*pERBB3^{7F}*)-transfected cells, NRG1β-induced p-AKT and p-ERK1/2 did not differ from wild-type human ERBB3 (*pERBB3*)-transfected cells. Following NRG1β induction, the molecular weight of the phosphorylated protein at the tyrosine residue was approximately 190-kD, which corresponded to the ERBB2 and ERBB3 in pERBB3-transfected cells. The level of phosphorylated protein at tyrosine residues was subdued in *pERBB3^{7F}*-transfected cells compared to *pERBB3*. *pERBB3^{7A}*-transfected cells particularly were more severely attenuated compared to others (Fig 6A).

After immunoprecipitation (IP) using anti-ERBB2 antibody or anti-ERBB3 antibody, western blot with anti-p-Tyr antibody was performed to identify the phosphorylation sites in

ERBB3 other than the p85 binding sites (Fig 6C). After treatment with NRG1β, the level of p-Tyr (phosphorylation of Tyrosine) after IP with anti-ERBB2 antibody was slightly increased only in the pERBB3-transfected cells. Similar to the previous p-Tyr blot results, p-Tyr expression in IP using an anti-ERBB3 antibody revealed the NRG1β-dependent elevation of p-ERBB3 in *pERBB3*$^{7F}$-transfected cells. However, the elevation of p-ERBB3 was lower than in *pERBB3*-transfected cells. In contrast, the basal and elevated p-ERBB3 was barely detected in *pERBB3*$^{7A}$-transfected cells. Overall, these data suggest that the tyrosine phosphorylation of the p85 binding sites in ERBB3 is predominant but not exclusive; NRG1β-dependent ERBB3 phosphorylation of other sites may also activate the AKT and ERK pathways.

## Conclusions

The anomalies during heart development observed in *Erbb3* mutants suggest the role of ERBB3 in developmental processes. ERBB3 is mainly expressed in the endocardial and mesenchymal cells of the AV cushion region at E9.5 to E10.5 during the EndMT. ERBB3 was only expressed in mesenchymal cells during E11.5 to E12.5 [10]. *Erbb3 null* and *7A* mutations caused embryonic lethality. In both mutations, mesenchymal cell proliferation was temporarily reduced at E10.5 and endocardial cell proliferation was significantly reduced at E11.5. In both mutants, the right ventricle was smaller than the left ventricle, and the myocardial thickness was reduced. Our results suggest that the heart defects in both the *Erbb3*$^{-/-}$ and *Erbb3*$^{7A/7A}$ mutants were caused by overall growth retardation.

VE-cadherin is specifically expressed in endocardial cells, and the regulation of VE-cadherin expression by Notch signaling directly affects EndMT [21–23]. As EndMT proceeds, the level of VE-cadherin is reduced in the endocardial cells, whereas the level of SNAIL is significantly increased. Our results suggest that the ERBB3-dependent signaling appears connected to the SNAIL-VE-cadherin pathway in regulating the EndMT for AV cushion morphogenesis. Also, the defects of *Erbb3* mutants have been implicated in the dysregulation of proliferation and EndMT during endocardial cushion development [20]. EndMT is essential for the development of the endocardial cushion. The previous report on ERBB2-ERBB3 knockouts showed that EndMT events were attenuated in heart explant cultures [10]. The EndMT of *Erbb3* mutants was also attenuated in our AVC-explant experiments.

The main pathways involved in Nrg1-ERBB3 signaling are PI3K and MAPK. Treatment with PI3K or MAPK inhibitors decreased EndMT in the heart explant culture. Moreover, the inhibition of PI3K and MAPK signaling dramatically reduced the transformation of endothelial cells to mesenchymal cells. In these results, both the PI3K and MAPK signals were essential for the EndMT event during the endocardial cushion formation, and *Erbb3* mutations inhibited EndMT due to the decline in PI3K and MAPK signaling (S4 Fig in S1 File). We additionally identified a reduction in the number of mesenchymal cells in the AV cushion due to failed EndMT event and decreased cell proliferation. These results suggest that the proliferation of mesenchymal cells in the AV cushion of *Erbb3* mutant embryos was transiently reduced due to attenuated AKT and ERK phosphorylation.

Previously, an *Erbb3* mutant with YXXM-to-FXXM mutations (*Erbb3*$^{7F}$) was analyzed in a mouse breast cancer model and revealed no embryonic lethality [11]. A possible explanation for this would be due to the differential NRG1-ERBB3 signaling between *Erbb3*$^{7A}$ and *Erbb3*$^{7F}$ mutations. NRG1-dependent p-AKT(T308) and p-AKT(S473) were clearly reduced in the ERBB3$^{7A}$, but not in ERBB3$^{7F}$, according to our results in CHO cells. ERBB3$^{7F}$ does not exhibit a decreased level of p-ERK1/2 contrary to ERBB3$^{7A}$, which is also likely to contribute to the survival of *Erbb3*$^{7F}$ embryos. Overall, our results suggest that the YXXA mutation blocks the

activation of ERBB3-dependent PI3K signaling more than does the FXXM mutation of ERBB3.

## Supporting information

**S1 File.**
(DOCX)

## Acknowledgments

We would like to thank Dr. Woong Sun for the helpful discussion.

## Author Contributions

**Conceptualization:** Kyoungmi Kim.

**Data curation:** Kyoungmi Kim.

**Formal analysis:** Kyoungmi Kim.

**Funding acquisition:** Daekee Lee.

**Investigation:** Daekee Lee.

**Methodology:** Daekee Lee.

**Project administration:** Daekee Lee.

**Supervision:** Daekee Lee.

**Validation:** Kyoungmi Kim.

**Visualization:** Kyoungmi Kim.

**Writing – original draft:** Kyoungmi Kim, Daekee Lee.

**Writing – review & editing:** Kyoungmi Kim, Daekee Lee.

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
