## [Decision Letter · Decision Letter 0]

23 Feb 2021

PONE-D-21-03408

ERBB3-dependent AKT and ERK pathways are essential for atrioventricular cushion development and valve formation in the mouse embryo

PLOS ONE

Dear Dr. Kim,

Thank you for submitting your manuscript to PLOS ONE. After careful consideration, we feel that it has merit but does not fully meet PLOS ONE’s publication criteria as it currently stands. Therefore, we invite you to submit a revised version of the manuscript that addresses the points raised during the review process.

Three excellent reviewers have read your manuscript and you will find their comments below. Please address the criticisms that the reviewers have raised in your resubmission and also make writing changes they have suggested.

We look forward to receiving your revised manuscript.

Kind regards,

Robert W Dettman, PhD

Academic Editor

PLOS ONE

Journal Requirements:

Reviewers' comments:

Reviewer's Responses to Questions

**Comments to the Author**

1. Is the manuscript technically sound, and do the data support the conclusions?

Reviewer #1: Partly

Reviewer #2: Partly

Reviewer #3: Partly

2. Has the statistical analysis been performed appropriately and rigorously? 

Reviewer #1: Yes

Reviewer #2: No

Reviewer #3: Yes

3. Have the authors made all data underlying the findings in their manuscript fully available?

Reviewer #1: Yes

Reviewer #2: Yes

Reviewer #3: Yes

4. Is the manuscript presented in an intelligible fashion and written in standard English?

Reviewer #1: No

Reviewer #2: Yes

Reviewer #3: No

5. Review Comments to the Author

Reviewer #1: This study by Kim et al., examines the endocardial cushion phenotype in mice harboring a mutant form of human ERBB3, whereby YXXM p85 binding sites were replaced with YXXA. Erbb3 knock-in embryos were embryonically lethal around E12.5-E13.5 with hypoplastic endocardial cushions, attributed to decreased transformation and proliferation; likely due to decreased pAkt and pErk. These findings are consistent with previous studies utilizing global Erbb3 null mice. The manuscript introduces a new, mouse model to the field that allows for studies focused on the role of ERBB3-dependent PI2K signaling during development. However, findings from the manuscript provide incremental insights into the field, and the data does now distinguish between cause, and effect of the ERBB3 mutation. In addition, without the inclusion of quantitative data, the conclusions might be considered overinterpreted.

1. Line 71 is not clear – what is meant by “ERBB3-dependent pathways are incorrectly activated for proliferation”?

2. Where is Erbb3 expressed during endocardial cushion development?

3. The cardiac phenotype of both Erbb3 mutants is very interesting. On line 218 the authors comment on a small right ventricle, and this is further expanded upon on lines 330-333, which include reports of “a small tricuspid valve, comparable thickness of the myocardium, thin myocardial wall, HRHS”. The authors should quantify these phenotypes based on histological sections; myocardial thickness, AV cushion size. In addition the penetrance of these phenotypes should be included.

4. Did the authors examine outflow tract valves?

5. A consistent concern throughout the manuscript is the number of studies and interpretations drawn from E13.5 embryos. The authors report 0% survivability by E14.5, and a significant loss at E13.5, and therefore there is a concern that phenotypes are being observed due to lethality and not directly related to the function of Erbb3 in these process. In Figure 2D, E, viability of the mutant embryos is clearly a problem as there is a lack of blood circulation.

6. Line 230-231, what is meant by AV cushions of mutant embryos were comparable with wild-type?

7. The authors comment several times on differences between endothelial, versus mesenchyme cells. How were the cell types determine?

8. Figure 3E is interesting. The quantitation of VE-cadherin and Snai1 is critical here, in addition to the number of long-shaped, versus rounded nuclei. The authors should also be cautious that the EMT defect may not be caused by changes in VE-cadherin and Snai1 expression, this may be a molecular readout of impaired EMT (and not the cause). There is certainly no evidence to suggest that the attenuation of EMT is directly related to a high level of VE-cadherin.

9. In Figure 5, the concern over embryo viability remains. Another point to consider is that valve elongation is not occurring due to the initial EMT defect. Following on, the reduced expression of Nfatc1 is likely attribute to survivability (or lack of), and EMT impairment and independent of ERBB3. The orientation of tissue sections in this figure are also not clear. Figure 5 is a major concern.

Reviewer #2: This article describes atrioventricular (AV) defects and deficient P13K/AKT and ERK pathways in Erbb3-null and knock-in mutant mouse embryos.

The authors provide detailed description of experimental procedures and present interesting observations of AV endocardial cushion/AV canal defects in Erbb3-null and knock-in mutant mouse embryos

Several major concerns are raised.

1. Page 13, Line 326- 328

The authors describe that ERBB3 is mainly expressed in the endocardial and mesenchymal cells in the AV cushion from E9.5 to E10 during AV endocardial mesenchymal transformation (EndMT) and ERBB3 is only expressed in mesenchymal cells from E11.5 to E12.5 by citing Camenisch et al., 2002 (ref. 10 in this paper). However, intense expression of ERBB3 is seen in AV myocardium in Figure 2 in Camenisch et al. (2002). Since the authors use conventional knock-in mice, the notion that ERBB3 is expressed both in endocardial and myocardial cells in the AV canal is critical to understand the nature of AV canal defects found in the mutant mice.

2. Page 9, Line 216-221 and Page 13, line 331-333.

The authors describe that both Erbb3-null and 7A mutant exhibit small right ventricle and AV septal defects at E12.5. However, cardiac septation is known to be established by E15.5 in the normal condition and therefore, defects found in the null and mutant hearts may not be thoroughly attributed to the AV septal defects. Because of the growth defects of the embryos seen at E13.5 (Fig.1D), these heart defects can be caused by overall growth retardation.

3. Page 9, Line 227-235.

The authors describe that Erbb3-null and 7A mutant have fewer mesenchymal cells in the AV endocardial cushion. However, at E11.5, the 7A mutant (Fig. 3A lower right panel) shows more mesenchymal cells in AV cushions as compared to the wild type (Fig. 3A lower left). The histogram (Fig. 3B) provided by the authors does not appear to represent actual cell density seen in Fig. 3A.

4. Page 10, Line 252-261.

The authors describe that proliferation ratio of mesenchymal cells is reduced in the null and 7A mutant at E10.5

If the mesenchymal cell proliferation ratio is decreased, that itself can directly cause reduction of cell density of mesenchymal cells in the AV cushion (Fig. 3B) without deficient endocardial cell transformation that they claimed in Fig. 3D. EndMT commences at around E9.5 but continues for a while until fusion of the superior and inferior AV cushions. Mesenchymal cell formation by EndMT and proliferation of mesenchymal cells occur in the same AV cushions at E10.5. Therefore, if there is significant reduction in proliferation of mesenchymal cells in the AV cushions at ED10.5, it is not possible to attribute reduction of cell density to reduction of the EndMT.

5. Page 11, Line 263-279 & Page 7, Line 172-178.

Authors claim that they collected AV endocardial cushions from E11.5 mouse embryos for protein extraction. However, at E11.5, it is technically very challenging to dissect out AV endocardial cushions without contamination with AV myocardium that associate with the AV cushion. Especially because the null and 7A mutants have smaller defective hearts, it is extremely difficult to correctly dissect out endocardial cushions.

The data presented here could likely represent protein extraction from E11.5 AV canal, that include endocardial cushions and associated myocardium. Also, it is important that AV myocardium intensely expresses ERBB3 (Camenisch et al., 2002, Ref. 10, Fig. 2). Therefore, the data obtained here would be largely coming from deficiencies of ERBB3 expression in the myocardium.

6. Page 11, line 281-Page 12, Line 239.

Here is the authors’ fundamental misunderstanding of AV valvulogenesis. AV valves are not directly formed from superior and inferior AV endocardial cushions. Superior and inferior AV endocardial cushions fuse in the midline of the heart to form the septal AV cushion and from the septal AV cushion, septal leaflets of AV valves are formed. Parietal leaflets of the AV valves are derived from lateral AV cushions not from superior and inferior AV cushions (Snarr et al., Dev Dyn 236, 1287-1294, 2007; Snarr et al., Dev Dyn 237, 2804-2819, 2008).

Therefore, arrows shown in Fig. 5A-B have nothing to do with the formation of AV valves. Figure 5C is showing fusion points of superior and inferior AV cushions.

The data presented here do not provide any evidence for interaction of NFATc1 and ERBB3 in AV valvulogenesis.

Other concerns:

The authors show histograms with different letters, a, ab and b. They indicate in the figure legends that the different letters on the bar graphs represent the statistically significant differences. It is not clear what was truly compared.

Reviewer #3: In this paper the authors analyze knock-in mice that express a mutant ERBB protein whose seven p85 binding sites have been altered to determine the role of ERBB3-dependent PI3K signaling in AV cushion morphogenesis.

1) For some readers the level of detail provided by the authors may be sufficient. However, as a relative novice to this particular protein and its downstream targets, I found it difficult to follow the reasoning of the authors and how they came to some of their conclusions. As a first step, I think it would be helpful for the authors to provide the reader with a clear understanding of the pathways discussed in this manuscript and how they intersect. Personally, I found it very hard to link some of the pathways being discussed and Figure S4 is not sufficiently detailed to provide much help.

2) The authors should also address the following concerns.

3) How do the authors account for the significant drop in expression of the the ERBB3 protein level in the Erbb37A/7A embryos to only one-third of that in the wild-type embryos? This significant drop could independently affect development. At a minimum, comparisons should be made with Erbb3+/- embryos whose expression level is similar to that of Erbb37A/7A embryos.

4) The data in Figure 1H should be quantified for comparison to the data in Figure 1G. Was this experiment repeated? What was the n (number of repeats)?

5) The interventicular spetum is not always complete at E12.5 in wild type embryos. In figure 2C the authors show what appears to be a complete septum in a wild type embryo, but this section does not correspond to the sections shown for the null and 7A/7A embryos. The authors should provide counts (how many embryos have complete septums) from all genotype and show comparable sections. Without this, it is very difficult to see whether there is a real difference at this stage.

6) The authors state that, “In contrast to the weak immunostaining of VE-cadherin between the endothelial and delaminated cells in wild-type embryos, the intense immunostaining of VE-cadherin between the cell types was detected more frequently in mutant embryos, indicating that the attenuation of EndMT is directly related to a high level of VE-cadherin.” The term directly related suggest causality which is not proved. These things may be correlated, but not causal. The authors will need to indicate that more clearly.

7) There appears to be a distinct difference in the level of SNAIL protein in the 7A/7A sections than in the -/- sections (figure 3E). Nuclei also look rounder and the VE-cadherin level looks different. What is the n of this experiment? Are these sections representative?

8) Supplemental figure 1 should be included in the manuscript since it contains important data.

9) For Figure 4A, the authors should provide graphs of AKT and ERK levels as well. Saying “they all had comparable levels of total protein” is ambiguous since we don’t know what that refers to

10) Figure 4B, no images are provided for E10.5, or did I miss them?

11) It is not obvious why the authors observation that “co-treatment of the AVC-explant with PI3K and MEK inhibitors achieved higher inhibition of the EndMT of the AV cushion than treatment with one inhibitor (Fig. S2A, B)” help us to conclude that, “the embryonic heart defect in the Erbb3 mutant embryos is likely due to the attenuation of both AKT and ERK phosphorylation.” If this does help us come to that conclusion, the authors need to help the reader to make that connection.

12) The authors state, “at E11.5 (Fig. S3B) and E12.5 (Fig. 5B, C), the nuclear localization of NFATC1 in endocardial cells where valve formation occurred was reduced in mutant embryos.” At E11.5, the images do not suggest that NFATC1 is in the cytoplasm of these cells (having failed to be localized to the nucleus). Rather that fewer cells are expressing NFATC1. Perhaps this is because we can’t see where the nuclei are in the double stained images. The authors should provide images with DAPI alone to help us visualize the nucleus of each cell.

13) The authors introduce the delta85 allele into the paper in the results section with no introduction (just a reference). What is this mutation? Why is it of interest?

14) If the authors wish to comment on the level of Nrg1β-induced p-AKT levels, they should normalize not only to AKT but to the Nrg1β negative levels. Without that normalization, it is very difficult to determine if there is attenuation of the Nrg1β response.

15) The authors state, “despite the absence of p-ERBB3 in pERBB37F309 -transfected cells, Nrg1β-induced p-AKT and p-ERK1/2 increased compared to that I the pERBB3-transfected cells.” Not sure what they meant by “…that I the…” Assuming they meant “….to that see in the…” it would not appear that this si the case for p-ERK whose response was similar to that of pERBB3 (a to c vs. a to c).

16) The authors state, “Erbb3 null and 7A mutations caused…thin myocardial wall.” Then they say that, “the thickness of the myocardium in the right and left ventricles was similar.” That is inconsistent.

17) The authors state, “the heart malformation of both Erbb3-/- and Erbb3331 7A/7A mutants was related with hypoplastic right heart syndrome (HRHS) and atrioventricular septal defect (AVSD).” This will require a better comparison of E12.5 heart sections in Figure 2C.

18) I don’t feel that the authors provided sufficient evidence to suggest that the “tricuspid valves of the mutants was smaller.

19) It would be inaccurate to say, “EndMT events failed in the heart explant culture” if you have any level of mesenchymal cell production. “Fail” suggest no EndMT. Words like, “reduced” or “attenuated” are a more accurate description of their data.

20) The authors show that their mutations are associated with decreases in p-AKT and p-ERK1/2 levels. I do not follow how this data and a supplemental experiment showing that PI3K and MEK inhibitors additively affect EndMT leads them to conclude that their “Erbb3 mutations inhibited EndMT due to the decline in… MAPK signaling (Fig. S4).” Perhaps I am just not familiar with these pathways, but the authors don’t provide enough explanation for me—a simple reader—to understand how they have proven an effect on MAPK signaling (which by the way is never mentioned in the abstract).

Minor

1) Proteins should be written in all capital letters.

2) The paper needs to be reviewed by a native English speaker.

3) The quality of the figures provided for review is very poor. It is really hard to see details. In revision, high resolution images should be submitted (300 dpi or higher).

6. PLOS authors have the option to publish the peer review history of their article (what does this mean?). If published, this will include your full peer review and any attached files.

Reviewer #1: No

Reviewer #2: No

Reviewer #3: No

---

## [Author Response · Author response to Decision Letter 0]

18 Aug 2021

Reviewer #1: 

This study by Kim et al., examines the endocardial cushion phenotype in mice harboring a mutant form of human ERBB3, whereby YXXM p85 binding sites were replaced with YXXA. Erbb3 knock-in embryos were embryonically lethal around E12.5-E13.5 with hypoplastic endocardial cushions, attributed to decreased transformation and proliferation; likely due to decreased pAkt and pErk. These findings are consistent with previous studies utilizing global Erbb3 null mice. The manuscript introduces a new, mouse model to the field that allows for studies focused on the role of ERBB3-dependent PI2K signaling during development. However, findings from the manuscript provide incremental insights into the field, and the data does now distinguish between cause, and effect of the ERBB3 mutation. In addition, without the inclusion of quantitative data, the conclusions might be considered overinterpreted.

1. Line 71 is not clear – what is meant by “ERBB3-dependent pathways are incorrectly activated for proliferation”?

Response: We agree that this line was not clear and did not fit appropriately in the context. Accordingly, we have deleted this sentence.

2. Where is Erbb3 expressed during endocardial cushion development?

Response: ERBB3 is expressed at E10.0 in the endocardial and mesenchymal cells in the AV cushion of the embryonic heart (Todd D. Camenisch et al., Nat Med., 2002). We also identified ERBB3 expression in the heart of an E10.5 embryo using ERBB3 antibody immunohistochemistry. At E10.5, ERBB3 is expressed only in the endocardial and mesenchymal cells in the AV cushion of the embryonic heart. These data can be found below, and have also been added to Fig. S1.

Figure S1. Histological ERBB3 expression in the heart of the wild-type E10.5. Section 1 represents the trabeculated myocardium, and section 2 represents the AV cushion.

3. The cardiac phenotype of both Erbb3 mutants is very interesting. On line 218 the authors comment on a small right ventricle, and this is further expanded upon on lines 330-333, which include reports of “a small tricuspid valve, comparable thickness of the myocardium, thin myocardial wall, HRHS”. The authors should quantify these phenotypes based on histological sections; myocardial thickness, AV cushion size. In addition the penetrance of these phenotypes should be included.

Response: Thank you for your comment. We have measured the myocardial thickness of E12.5 embryos. Both Erbb3-/- and Erbb37A/7A had reduced myocardial wall thickness compared to the wild type. These data can be found below and have been added to Fig. S4.

In Figure 3C, we had previously measured the number of mesenchymal cells in the AV cushion. These data showed that the number of mesenchymal cells in the AV cushion was reduced in both mutant embryos. This reduction of mesenchymal cells is expected to eventually affect the size of the AV cushion. 

Figure. S4. Myocardial wall thickness of E12.5 embryos. (A) At E12.5, the myocardial wall of the heart was examined by hematoxylin and eosin (H&E) staining. Scale bar = 50 µm. (B) Quantification of myocardial wall thickness in each samples of Erbb3+/+, Erbb3-/-, and Erbb37A/7A.

4. Did the authors examine outflow tract valves?

Response: We focused only on AV cushion formation.

5. A consistent concern throughout the manuscript is the number of studies and interpretations drawn from E13.5 embryos. The authors report 0% survivability by E14.5, and a significant loss at E13.5, and therefore there is a concern that phenotypes are being observed due to lethality and not directly related to the function of Erbb3 in these process. In Figure 2D, E, viability of the mutant embryos is clearly a problem as there is a lack of blood circulation.

Response: We agree with your opinion. Most phenotypes were analyzed at the E10.5 to E12.5 stages before lethality occurred. The E13.5 embryo phenotypes were investigated to determine the direct reason for the lethality. 

6. Line 230-231, what is meant by AV cushions of mutant embryos were comparable with wild-type?

Response: We corrected the phrase on p.9, line 234-235, as follows: “In E10.5 embryos of Erbb37A/7A and Erbb3-/-, the number of mesenchymal cells in the AV cushion was reduced compared to the wild type (Fig. 3A, B).” We have also marked this sentence with a yellow highlight in the manuscript.

7. The authors comment several times on differences between endothelial, versus mesenchyme cells. How were the cell types determine?

Response: Todd D. Camenisch et al. (Nat Med., 2002/reference 10) and Ching-Pin Chang et al. (Cell, 2004/reference 20) showed EMT analysis using the E9.0 endocardial cushion to study the role of calcineurin signaling during the transformation process. We distinguished the difference between endothelial and mesenchymal cells by referring to the findings of these studies. 

8. Figure 3E is interesting. The quantitation of VE-cadherin and Snai1 is critical here, in addition to the number of long-shaped, versus rounded nuclei. The authors should also be cautious that the EMT defect may not be caused by changes in VE-cadherin and Snai1 expression, this may be a molecular readout of impaired EMT (and not the cause). There is certainly no evidence to suggest that the attenuation of EMT is directly related to a high level of VE-cadherin.

Response: We tried but were unable to obtain ERBB3 mutant embryos for quantification of VE-cadherin and SNAIL during the revision period. We agree with your opinion that alterations in VE-cadherin and Snai1 expression may not be the cause of impaired EndMT. However, many previous studies reported that the regulation of SNAIL-VE-cadherin expression has a direct effect on EndMT (Luika A. Timmerman et al., Genes Dev. 2004; Sonsoles Piera-Velazquez et al., Physiol Rev. 2019). We found that whereas EndMT progression reduced the level of VE-cadherin in the endocardial cells, it significantly increased the level of Snai1 as an EndMT transcription factor. Overall, ERBB3-dependent signaling during AV cushion morphogenesis appears to be connected to the SNAIL-VE-cadherin pathway for regulating EndMT.

9. In Figure 5, the concern over embryo viability remains. Another point to consider is that valve elongation is not occurring due to the initial EMT defect. Following on, the reduced expression of Nfatc1 is likely attribute to survivability (or lack of), and EMT impairment and independent of ERBB3. The orientation of tissue sections in this figure are also not clear. Figure 5 is a major concern.

Response: We conducted experiments using only live E12.5 embryos. We agree that valve elongation did not occur due to the initial EMT defect. These heart defects can be caused by overall growth retardation. We also identified that AV valves are not directly formed from the superior and inferior AV endocardial cushions. Rather, the superior and inferior AV endocardial cushions fuse in the heart’s midline to form the septal AV cushion and septal leaflets of the AV valves. The parietal leaflets of the AV valves are derived from the lateral AV cushions, and not from the superior and inferior AV cushions (Brian S. Snarr et al., Dev Dyn. 2007; Brian S. Snarr et al., Dev Dyn. 2008). Therefore, Figure 5 is in error and has been removed from the manuscript.

Reviewer #2: 

This article describes atrioventricular (AV) defects and deficient P13K/AKT and ERK pathways in Erbb3-null and knock-in mutant mouse embryos.

The authors provide detailed description of experimental procedures and present interesting observations of AV endocardial cushion/AV canal defects in Erbb3-null and knock-in mutant mouse embryos

Several major concerns are raised.

1. Page 13, Line 326- 328

The authors describe that ERBB3 is mainly expressed in the endocardial and mesenchymal cells in the AV cushion from E9.5 to E10 during AV endocardial mesenchymal transformation (EndMT) and ERBB3 is only expressed in mesenchymal cells from E11.5 to E12.5 by citing Camenisch et al., 2002 (ref. 10 in this paper). However, intense expression of ERBB3 is seen in AV myocardium in Figure 2 in Camenisch et al. (2002). Since the authors use conventional knock-in mice, the notion that ERBB3 is expressed both in endocardial and myocardial cells in the AV canal is critical to understand the nature of AV canal defects found in the mutant mice.

Response: Todd D Camenisch et al., (Nat Med., 2002) noted as follows: “ErbB3 is present in the invading mesenchyme and endocardium. Although ErbB3 is detected in myocardium, it may be nonspecific as some staining is observed in the myocardium of ErbB3–/– embryos.” In conclusion, ERBB3 is expressed only in the endocardial and mesenchymal cells in the embryonic heart’s AV cushion at E10.0. 

We also identified ERBB3 expression in the heart of E10.5 embryos using ERBB3 antibody immunohistochemistry. ERBB3 was expressed only in the endocardial and mesenchymal cells in the embryonic heart AV cushion at E10.5. These data can be found below and have been added to Fig. S1.

Figure S1. Histological ERBB3 expression in the heart of the wild-type E10.5. Section 1 represents the trabeculated myocardium, and section 2 represents the AV cushion.

2. Page 9, Line 216-221 and Page 13, line 331-333.

The authors describe that both Erbb3-null and 7A mutant exhibit small right ventricle and AV septal defects at E12.5. However, cardiac septation is known to be established by E15.5 in the normal condition and therefore, defects found in the null and mutant hearts may not be thoroughly attributed to the AV septal defects. Because of the growth defects of the embryos seen at E13.5 (Fig.1D), these heart defects can be caused by overall growth retardation.

Response: We agree with your opinion and have modified the phrase on p.13, lines 327-329, as follows: “Overall, our results suggest that heart defects of both Erbb3-/- and Erbb37A/7A mutants were caused by overall growth retardation.” We have also marked this sentence with a yellow highlight in the manuscript.

3. Page 9, Line 227-235.

The authors describe that Erbb3-null and 7A mutant have fewer mesenchymal cells in the AV endocardial cushion. However, at E11.5, the 7A mutant (Fig. 3A lower right panel) shows more mesenchymal cells in AV cushions as compared to the wild type (Fig. 3A lower left). The histogram (Fig. 3B) provided by the authors does not appear to represent actual cell density seen in Fig. 3A.

Response: We quantified the number of mesenchymal cells in the AV cushion up to n=8 in E11.5-stage embryos. Our results identified that the number of mesenchymal cells differed significantly between Erbb3 mutants and the wild type. 

4. Page 10, Line 252-261.

The authors describe that proliferation ratio of mesenchymal cells is reduced in the null and 7A mutant at E10.5. If the mesenchymal cell proliferation ratio is decreased, that itself can directly cause reduction of cell density of mesenchymal cells in the AV cushion (Fig. 3B) without deficient endocardial cell transformation that they claimed in Fig. 3D. EndMT commences at around E9.5 but continues for a while until fusion of the superior and inferior AV cushions. Mesenchymal cell formation by EndMT and proliferation of mesenchymal cells occur in the same AV cushions at E10.5. Therefore, if there is significant reduction in proliferation of mesenchymal cells in the AV cushions at ED10.5, it is not possible to attribute reduction of cell density to reduction of the EndMT.

Response: Thank you for your comment. We agree that the decreased mesenchymal cell proliferation rate may itself decrease the mesenchymal cell density in the AV cushion. However, we found a low number of mesenchymal cells in the AV cushion at E10.5. This result suggests that the formation of mesenchymal cells by early EndMT was also defective. Overall, the defects in the AV cushion are due both to the lack of mesenchymal cell formation by early EndMT, and the decreased proliferation of mesenchymal cells in the AV cushion.

5. Page 11, Line 263-279 & Page 7, Line 172-178.

Authors claim that they collected AV endocardial cushions from E11.5 mouse embryos for protein extraction. However, at E11.5, it is technically very challenging to dissect out AV endocardial cushions without contamination with AV myocardium that associate with the AV cushion. Especially because the null and 7A mutants have smaller defective hearts, it is extremely difficult to correctly dissect out endocardial cushions.

The data presented here could likely represent protein extraction from E11.5 AV canal, that include endocardial cushions and associated myocardium. Also, it is important that AV myocardium intensely expresses ERBB3 (Camenisch et al., 2002, Ref. 10, Fig. 2). Therefore, the data obtained here would be largely coming from deficiencies of ERBB3 expression in the myocardium.

Response: We carefully dissected the heart’s AVC region at the E11.5 stage using fine iris scissors on a dissecting microscope. We cannot completely exclude the possibility of contamination with AV myocardium associated with the AV cushion. However, as mentioned earlier, ERBB3 is expressed in the endocardial and mesenchymal cells in the AV cushion of the embryonic heart. Therefore, the data obtained here would largely come from deficiencies of ERBB3 expression in the endocardial and mesenchymal cells in the embryonic heart’s AV cushion.

6. Page 11, line 281-Page 12, Line 239.

Here is the authors’ fundamental misunderstanding of AV valvulogenesis. AV valves are not directly formed from superior and inferior AV endocardial cushions. Superior and inferior AV endocardial cushions fuse in the midline of the heart to form the septal AV cushion and from the septal AV cushion, septal leaflets of AV valves are formed. Parietal leaflets of the AV valves are derived from lateral AV cushions not from superior and inferior AV cushions (Snarr et al., Dev Dyn 236, 1287-1294, 2007; Snarr et al., Dev Dyn 237, 2804-2819, 2008). Therefore, arrows shown in Fig. 5A-B have nothing to do with the formation of AV valves. Figure 5C is showing fusion points of superior and inferior AV cushions. The data presented here do not provide any evidence for interaction of NFATc1 and ERBB3 in AV valvulogenesis.

Response: Thank you very much for your comments. We agree that AV valves are not directly formed from the superior and inferior AV endocardial cushions. Therefore, Figure 5 is in error and has been removed from the manuscript. We have accordingly revised the manuscript and figures for AV valvulogenesis.

Other concerns:

The authors show histograms with different letters, a, ab and b. 

They indicate in the figure legends that the different letters on the bar graphs represent the statistically significant differences. It is not clear what was truly compared.

Response: We revised the phrase on p.8, lines 195-199, as follows: “Each letter (a, b, c or d) in the graph indicates a significant difference between experimental groups. Two letters that are the same (e.g., a and a) indicate a nonsignificant difference (ns), whereas different letters (e.g., a and b) indicate a significant difference. Two letters together (e.g., ab) indicates non-significant differences (ns) with either a or b, but a significant difference compared with c or d.” We have also marked this revised phrase with yellow highlighting in the manuscript.

Reviewer #3: 

In this paper the authors analyze knock-in mice that express a mutant ERBB protein whose seven p85 binding sites have been altered to determine the role of ERBB3-dependent PI3K signaling in AV cushion morphogenesis.

1) For some readers the level of detail provided by the authors may be sufficient. However, as a relative novice to this particular protein and its downstream targets, I found it difficult to follow the reasoning of the authors and how they came to some of their conclusions. As a first step, I think it would be helpful for the authors to provide the reader with a clear understanding of the pathways discussed in this manuscript and how they intersect. Personally, I found it very hard to link some of the pathways being discussed and Figure S4 is not sufficiently detailed to provide much help. 

Response: We agree with your opinion. We admit that there is a lack of direct evidence to explain Figure S4. As such, Figure S4 has been removed and its details have been added to the manuscript’s conclusion. 

2) The authors should also address the following concerns.

3) How do the authors account for the significant drop in expression of the the ERBB3 protein level in the Erbb37A/7A embryos to only one-third of that in the wild-type embryos? This significant drop could independently affect development. At a minimum, comparisons should be made with Erbb3+/- embryos whose expression level is similar to that of Erbb37A/7A embryos.

Response: We agree with your opinion. In Figure 1G, we confirmed that ERBB3 protein expression was similar in Erbb3+/- and Erbb37A/7A. For this reason, we also compared Erbb3+/- embryos in most experiments. However, Erbb3+/- showed a phenotype close to Erbb3+/+ (wild type).

4) The data in Figure 1H should be quantified for comparison to the data in Figure 1G. Was this experiment repeated? What was the n (number of repeats)?

Response: This experiment was repeated three times. Herein, we have quantified the western blot data and added it to Figure 1H. 

5) The interventicular spetum is not always complete at E12.5 in wild type embryos. In figure 2C the authors show what appears to be a complete septum in a wild type embryo, but this section does not correspond to the sections shown for the null and 7A/7A embryos. The authors should provide counts (how many embryos have complete septums) from all genotype and show comparable sections. Without this, it is very difficult to see whether there is a real difference at this stage.

Response: Under normal conditions, cardiac septation is known to be established by E15.5. Therefore, the defects found in the null and mutant hearts may not be thoroughly attributed to AV septal defects. These heart defects may be caused by overall growth retardation owing to the embryo growth defects seen at E13.5 (Fig. 1D). 

6) The authors state that, “In contrast to the weak immunostaining of VE-cadherin between the endothelial and delaminated cells in wild-type embryos, the intense immunostaining of VE-cadherin between the cell types was detected more frequently in mutant embryos, indicating that the attenuation of EndMT is directly related to a high level of VE-cadherin.” The term directly related suggest causality which is not proved. These things may be correlated, but not causal. The authors will need to indicate that more clearly.

Response: We agree with your opinion. We have modified the phrase on p.10, lines 247-250, as follows: “In contrast to VE-cadherin’s weak immunostaining between the endothelial and delaminated cells in wild-type embryos, we more frequently detected intense VE-cadherin immunostaining between the cell types in mutant embryos. These results suggest the possibility that attenuation of EndMT is related to high VE-cadherin levels (Fig. 3E).” We have also marked this passage with a yellow highlight in the manuscript.

7) There appears to be a distinct difference in the level of SNAIL protein in the 7A/7A sections than in the -/- sections (figure 3E). Nuclei also look rounder and the VE-cadherin level looks different. What is the n of this experiment? Are these sections representative?

Response: The sample size used in the experiment was n=3, and this figure is a representative image. 

8) Supplemental figure 1 should be included in the manuscript since it contains important data.

Response: We have moved these data from Supplemental figure 1 to main Figure 4.

9) For Figure 4A, the authors should provide graphs of AKT and ERK levels as well. Saying “they all had comparable levels of total protein” is ambiguous since we don’t know what that refers to 

Response: The quantified AKT and ERK data were added to Fig. S2 A and B. We also corrected the phrase on p.11, line 269-271, as follows: “The relative levels of phosphorylated AKT (p-AKT(S473)) and ERK1/2(p-ERK1/2) were lower in both mutant embryos than in the wild type, although their total protein levels did not differ (Fig. 5A, S2).” We have also marked this sentence with a yellow highlight in the manuscript.

10) Figure 4B, no images are provided for E10.5, or did I miss them?

Response: It is correct that there is no image via immunohistochemistry staining of E10.5. We performed immunofluorescence staining in E10.5 and E11.5 embryos to measure the relative intensities of the P-AKT and P-ERK proteins. Immunohistochemistry staining was performed and imaged only at the representative E11.5 stage. 

11) It is not obvious why the authors observation that “co-treatment of the AVC-explant with PI3K and MEK inhibitors achieved higher inhibition of the EndMT of the AV cushion than treatment with one inhibitor (Fig. S2A, B)” help us to conclude that, “the embryonic heart defect in the Erbb3 mutant embryos is likely due to the attenuation of both AKT and ERK phosphorylation.” If this does help us come to that conclusion, the authors need to help the reader to make that connection.

Response: We agree with your opinion. We have added the phrase on p.11, lines 278-284, as follows: “We tested co-treatment of the AVC explant with PI3K and MEK inhibitors to determine whether the reduced expression of the PI3K and MEK pathways by ERBB37A mutation affected early EndMT. The co-treatment of the AVC explant with PI3K and MEK inhibitors achieved higher EndMT inhibition within the AV cushion than did treatment with one inhibitor (Fig. S3A, B). These results indicate that the embryonic heart defect in Erbb3 mutant embryos was due to the reduction in early EndMT and attenuation of both AKT and ERK phosphorylation.” We have marked this passage with a yellow highlight in the manuscript.

12) The authors state, “at E11.5 (Fig. S3B) and E12.5 (Fig. 5B, C), the nuclear localization of NFATC1 in endocardial cells where valve formation occurred was reduced in mutant embryos.” At E11.5, the images do not suggest that NFATC1 is in the cytoplasm of these cells (having failed to be localized to the nucleus). Rather that fewer cells are expressing NFATC1. Perhaps this is because we can’t see where the nuclei are in the double stained images. The authors should provide images with DAPI alone to help us visualize the nucleus of each cell.

Response: We agree with your opinion. The data presented here do not provide any evidence for the interaction of NFATc1 and ERBB3 in AV valvulogenesis. Accordingly, we have removed Fig. S3B and Fig. 5C.

13) The authors introduce the delta85 allele into the paper in the results section with no introduction (just a reference). What is this mutation? Why is it of interest?

Response: The ErbB3△85 allele refers to a mouse in which the ERBB3 and PI3K pathways were isolated from YXXM to FXXM by replacing the Erbb3 gene with human ERBB3. Tyrosine-encoding codons at the positions 941, 1054, 1197, 1222, 1260, 1276, and 1289 within the YXXM PI3K p85 consensus binding sites were mutated at the second position (A to T) into phenylalanine-encoding codons. To reflect this information, we have added the phrase on p.11, lines 287-288, as follows: “Lahlou H et al. reported the generation of a Erbb3Δ85 knock-in mouse, in which the seven YXXM PI3K p85 consensus binding sites of human ERBB3 were replaced with FXXM.” We have also marked this sentence with a yellow highlight in the manuscript.

14) If the authors wish to comment on the level of Nrg1β-induced p-AKT levels, they should normalize not only to AKT but to the Nrg1β negative levels. Without that normalization, it is very difficult to determine if there is attenuation of the Nrg1β response.

Response: Thank you for your comment. However, this opinion has been sufficiently confirmed in the Figure 7 data.

15) The authors state, “despite the absence of p-ERBB3 in pERBB37F309 -transfected cells, Nrg1β-induced p-AKT and p-ERK1/2 increased compared to that I the pERBB3-transfected cells.” Not sure what they meant by “…that I the…” Assuming they meant “….to that see in the…” it would not appear that this si the case for p-ERK whose response was similar to that of pERBB3 (a to c vs. a to c).

Response: we revised the phrase on p.12, lines 301-304, as follows: “Despite the absence of phosphorylated ERBB3 (p-ERBB3) in FXXM mutation human ERBB3 (pERBB37F)-transfected cells, NRG1β-induced p-AKT and p-ERK1/2 did not differ from wild-type human ERBB3 (pERBB3)-transfected cells.” We have also marked this sentence with a yellow highlight in the manuscript.

16) The authors state, “Erbb3 null and 7A mutations caused…thin myocardial wall.” Then they say that, “the thickness of the myocardium in the right and left ventricles was similar.” That is inconsistent.

Response: We agree with your opinion. We have revised the phrase on p.13, lines 324-328, as follows: “The Erbb3 null and 7A mutations caused embryonic lethality. In both mutations, mesenchymal cell proliferation was temporarily reduced at E10.5 and endocardial cell proliferation was significantly reduced at E11.5. In both mutants, the right ventricle was smaller than the left ventricle, and the myocardial thickness was reduced. Our results suggest that the heart defects in both the Erbb3-/- and Erbb37A/7A mutants were caused by overall growth retardation.” We have also marked this passage with a yellow highlight in the manuscript.

17) The authors state, “the heart malformation of both Erbb3-/- and Erbb37A/7A mutants was related with hypoplastic right heart syndrome (HRHS) and atrioventricular septal defect 

(AVSD).” This will require a better comparison of E12.5 heart sections in Figure 2C.

Response: We judged that this sentence was over-interpreted in the conclusion. Cardiac septation is known to be established by E15.5 in the normal condition, and therefore, defects found in the null and mutant hearts may not be thoroughly attributed to AV septal defects. These heart defects may be caused by overall growth retardation due to the growth defects in the E13.5 embryos (Fig. 1D). Hence, we have revised the phrase on p.13, lines 327-328, as follows: “Our results suggest that the heart defects in both the Erbb3-/- and Erbb37A/7A mutants were caused by overall growth retardation.” We have also marked this sentence with a yellow highlight in the manuscript.

18) I don’t feel that the authors provided sufficient evidence to suggest that the “tricuspid valves of the mutants was smaller.

Response: We agree with your opinion. We identified that AV valves are not directly formed from the superior and inferior AV endocardial cushions. Rather, the superior and inferior AV endocardial cushions fuse in the heart’s midline to form the septal AV cushion and septal leaflets of the AV valves. The parietal leaflets of the AV valves are derived from the lateral AV cushions, and not from the superior and inferior AV cushions (Brian S. Snarr et al., Dev Dyn. 2007; Brian S. Snarr et al., Dev Dyn. 2008). Therefore, Figure 5 is in error and has been removed from the manuscript.

19) It would be inaccurate to say, “EndMT events failed in the heart explant culture” if you have any level of mesenchymal cell production. “Fail” suggest no EndMT. Words like, “reduced” or “attenuated” are a more accurate description of their data.

Response: We agree with your opinion. We corrected the phrase on p.13, lines 335-337, as follows: “The previous report on ERBB2-ERBB3 knockouts showed that EndMT events were attenuated in heart explant cultures. The EndMT of Erbb3 mutants was also attenuated in our AVC-explant experiments.” We have also marked this passage with a yellow highlight in the manuscript.

20) The authors show that their mutations are associated with decreases in p-AKT and p-ERK1/2 levels. I do not follow how this data and a supplemental experiment showing that PI3K and MEK inhibitors additively affect EndMT leads them to conclude that their “Erbb3 mutations inhibited EndMT due to the decline in… MAPK signaling (Fig. S4).” Perhaps I am just not familiar with these pathways, but the authors don’t provide enough explanation for me—a simple reader—to understand how they have proven an effect on MAPK signaling (which by the way is never mentioned in the abstract).

Response: We agree with your opinion. We have added the phrase on p.13-14, lines 338-346, as follows: “The main pathways involved in Nrg1-ERBB3 signaling are PI3K and MAPK. Treatment with PI3K or MAPK inhibitors decreased EndMT in the heart explant culture. Moreover, the inhibition of PI3K and MAPK signaling dramatically reduced the transformation of endothelial cells to mesenchymal cells. In these results, both the PI3K and MAPK signals were essential for the EndMT event during the endocardial cushion formation, and Erbb3 mutations inhibited EndMT due to the decline in PI3K and MAPK signaling (Fig. S4). We additionally identified a reduction in the number of mesenchymal cells in the AV cushion due to failed EndMT event and decreased cell proliferation. These results suggest that the proliferation of mesenchymal cells in the AV cushion of Erbb3 mutant embryos was transiently reduced due to attenuated AKT and ERK phosphorylation.” We have also marked this passage with a yellow highlight in the manuscript. 

Minor

1) Proteins should be written in all capital letters.

Response: We corrected all protein names to capital letters (Snail to SNAIL).

2) The paper needs to be reviewed by a native English speaker.

Response: Our manuscript was revised by a professional native English editor.

3) The quality of the figures provided for review is very poor. It is really hard to see details. In revision, high resolution images should be submitted (300 dpi or higher).

Response: We changed our figures to 300 dpi to increase the resolution.

---

## [Decision Letter · Decision Letter 1]

22 Sep 2021

PONE-D-21-03408R1ERBB3-dependent AKT and ERK pathways are essential for atrioventricular cushion development in mouse embryosPLOS ONE

Dear Dr. Kim,

Thank you for submitting your manuscript to PLOS ONE. After careful consideration, we feel that it has merit but does not fully meet PLOS ONE’s publication criteria as it currently stands. Therefore, we invite you to submit a revised version of the manuscript that addresses the points raised during the review process.

 The reviewers have recommended minor revisions, so if the recommended changes can be made I will make a decision without sending the manuscript back to the reviewers. Please ensure that your decision is justified on PLOS ONE’s publication criteria and not, for example, on novelty or perceived impact.

We look forward to receiving your revised manuscript.

Kind regards,

Robert W Dettman, PhD

Academic Editor

PLOS ONE

Journal Requirements:

Additional Editor Comments (if provided):

Reviewers' comments:

Reviewer's Responses to Questions

**Comments to the Author**

1. If the authors have adequately addressed your comments raised in a previous round of review and you feel that this manuscript is now acceptable for publication, you may indicate that here to bypass the “Comments to the Author” section, enter your conflict of interest statement in the “Confidential to Editor” section, and submit your "Accept" recommendation.

Reviewer #2: (No Response)

Reviewer #3: (No Response)

2. Is the manuscript technically sound, and do the data support the conclusions?

Reviewer #2: Partly

Reviewer #3: Partly

3. Has the statistical analysis been performed appropriately and rigorously? 

Reviewer #2: Yes

Reviewer #3: Yes

4. Have the authors made all data underlying the findings in their manuscript fully available?

Reviewer #2: Yes

Reviewer #3: Yes

5. Is the manuscript presented in an intelligible fashion and written in standard English?

Reviewer #2: Yes

Reviewer #3: Yes

6. Review Comments to the Author

Reviewer #2: The authors addressed several concerns raised by this reviewer. Specifically, the authors admit their misunderstanding of AV valve morphogenesis and corrected their statements in the manuscript and removed related-figures. The authors also corrected their statements for AV septal morphogenesis.

However, there are some other concerns remained to be addressed.

The authors also agreed that low cell density in the mutant AV cushions can be explained by the low proliferation ratio of endocardial mesenchymal cells in the mutant AV cushions without having EndMT failure.

The AV explants were dissected out from the embryos at E9.5 and cultured on the collagen gels for 48 hours. The authors show a decrease in mesenchymal cell numbers in the mutant AV cushions cultured on the collagen gels (Fig. 3D). During the 48 hours, newly formed mesenchymal cells proliferate. Proliferation ratio of mesenchymal cells in mutant mice is significantly lower than that of wild type mesenchymal cells (Fig. 4B) at E10.5. Lower cell numbers of mesenchymal cells derived from mutant AV explants and lower cell densities in mutant AV cushion mesenchyme can be simply explained by lower proliferation ratio of mutant AV cushion mesenchymal cells.

In case cell numbers and densities are low but there is no significant difference in cell proliferation ratio between mutant and wild type AV cushion mesenchymal cells, one can claim that there is EndMT defect.

In this paper, the authors do not present clear evidence that there are EndMT defects in the mutant AV cushions. The data the authors present in this manuscript show that the mutant mice exhibit defects in AV endocardial cushion development. The abstract should be revised to reflect their findings.

Abstract, Page 2, Line 33, Erbb3 knock-in-embryos showed a decrease in the endocardial mesenchymal transformation (EndMT)

The following description is suggested:

Erbb3 knock-in-embryos showed a decrease in mesenchymal cell numbers and density in AV cushions.

Minor issue:

As the reviewer #3 suggests, the authors have to improve qualities of their figures. According to the figures this reviewer can see through the web, figures appear still very vague and not sharp enough for actual publication.

Reviewer #3: I appreciate the authors’ efforts to address the changes requested by the reviewers. I have read the new manuscript and there are several issues that still need to be addressed.

1) The sentence, “were caused by overall (Fig. 2C).” on line 223 appears to have been truncated.

2) Remove the word “Expected” in Figure 2A

3) If the number of mesenchymal cells is lower in WT and 7a/7a embryos, but the density is the same, then that indicates that the endocardial cushion size in 7A/7A embryos is smaller. Correct?

4) The authors state that, VE-cadherin is a “key player” in EMT. They should state specifically its role in EMT. I assume that in this context, VE-cadherin represses EMT. Correct?

5) For Fig 5A it appears that the authors selected differnet p-AKT and AKT lanes for their analysis of wild-type, heterozygous and null embryos. The GAPDH control lanes were also selected form alternative lanes as well (see Supplemental Material). This is unacceptable.

6) Fig. 5B. Show the E10.5 immunohistochemistry.

7) Fig. 5C. Show the E10.5 immunohistochemistry.

8) I agree that the authors show that, “In both mutants, the right ventricle was smaller than the left ventricle, and the myocardial thickness was reduced.” I however, disagree that these results, “suggest that the heart defects in both the Erbb3-/- and Erbb37A/7A mutants were caused by overall growth retardation.” If that were true, then you would expect an overall reduction in heart size, not in defects in only one ventricle. They may not have a good explanation, but no explanation is better than one that can’t be supported by the data.

9) I also note that the change in the right ventricle size compared to the left ventricle is easy to see at E12.5, The authors show no embryo photos at that stage. Instead, the small embryo size is seen at E13.5 when the hearts in section also seem to be overall smaller than wild-type.

10) Was the myocardial thickness of the left ventricle affected?

11) The authors state, “Overall, our results suggest that the YXXA mutation completely blocks the activation of ERBB3-dependent PI3K signaling than does the FXXM mutation of ERBB3.” This is not grammatically correct. Did they mean to say, “Overall, our results suggest that the YXXA mutation blocks the activation of ERBB3-dependent PI3K signaling more so than does the FXXM mutation of ERBB3.”

12) The authors also state, “Furthermore, our data suggest that ERBB3-dependent PI3K signaling is essential for AV cushion development and valve formation.” The AV cushions develop in ERBB3 null mice, right? Hence, ERRB3 can’t be essential for AV cushion development. In addition, I have seen no evidence of any effect on valve formation in this paper.

7. PLOS authors have the option to publish the peer review history of their article (what does this mean?). If published, this will include your full peer review and any attached files.

Reviewer #2: No

Reviewer #3: No

---

## [Author Response · Author response to Decision Letter 1]

6 Oct 2021

Point-by-point Responses 

We thank the reviewers for their positive and constructive comments and suggestions. In response to their comments, we have accordingly revised and rearranged the manuscript and figures. Please find our point-by-point responses to the reviewers’ constructive criticisms below.

Reviewers' comments:

Reviewer #2:

The authors addressed several concerns raised by this reviewer. Specifically, the authors admit their misunderstanding of AV valve morphogenesis and corrected their statements in the manuscript and removed related-figures. The authors also corrected their statements for AV septal morphogenesis. However, there are some other concerns remained to be addressed. The authors also agreed that low cell density in the mutant AV cushions can be explained by the low proliferation ratio of endocardial mesenchymal cells in the mutant AV cushions without having EndMT failure.

The AV explants were dissected out from the embryos at E9.5 and cultured on the collagen gels for 48 hours. The authors show a decrease in mesenchymal cell numbers in the mutant AV cushions cultured on the collagen gels (Fig. 3D). During the 48 hours, newly formed mesenchymal cells proliferate. Proliferation ratio of mesenchymal cells in mutant mice is significantly lower than that of wild type mesenchymal cells (Fig. 4B) at E10.5. Lower cell numbers of mesenchymal cells derived from mutant AV explants and lower cell densities in mutant AV cushion mesenchyme can be simply explained by lower proliferation ratio of mutant AV cushion mesenchymal cells.

In case cell numbers and densities are low but there is no significant difference in cell proliferation ratio between mutant and wild type AV cushion mesenchymal cells, one can claim that there is EndMT defect. In this paper, the authors do not present clear evidence that there are EndMT defects in the mutant AV cushions. The data the authors present in this manuscript show that the mutant mice exhibit defects in AV endocardial cushion development. The abstract should be revised to reflect their findings. Abstract, Page 2, Line 33, Erbb3 knock-in-embryos showed a decrease in the endocardial mesenchymal transformation (EndMT).

The following description is suggested: Erbb3 knock-in-embryos showed a decrease in mesenchymal cell numbers and density in AV cushions.

Response: We agree with your proposed suggestion and have modified the phrase on p.2, lines 32–34, as follows: “Erbb3 knock-in embryos exhibited lethality between E12.5 to E13.5, and showed a decrease in mesenchymal cell numbers and density in AV cushions.”

Minor issue:

As the reviewer #3 suggests, the authors have to improve qualities of their figures. According to the figures this reviewer can see through the web, figures appear still very vague and not sharp enough for actual publication.

Response: Thank you for the helpful feedback. We replaced the original figures with revised figures using PACE to improve their quality.

Reviewer #3:

I appreciate the authors’ efforts to address the changes requested by the reviewers. I have read the new manuscript and there are several issues that still need to be addressed.

1) The sentence, “were caused by overall (Fig. 2C).” on line 223 appears to have been truncated.

Response: Our apologies for the unintended omission. We have modified the sentence on p.9, lines 221–222, as follows: “Microscopic and histological examination results of E12.5 embryos showed that both Erbb37A/7A and Erbb3-/- embryos have a small right ventricle (Fig. 2B, C).”

2) Remove the word “Expected” in Figure 2A

Response: We have removed the word "Expected" from Figure 2A.

3) If the number of mesenchymal cells is lower in WT and 7a/7a embryos, but the density is the same, then that indicates that the endocardial cushion size in 7A/7A embryos is smaller. Correct?

Response: Yes, that is correct.

4) The authors state that, VE-cadherin is a “key player” in EMT. They should state specifically its role in EMT. I assume that in this context, VE-cadherin represses EMT. Correct?

Response: Thank you for your query. Midgett et al. (Frontiers in Physiology, 2017, doi: 10.3389/fphys.2017.00056.) reported that VE-cadherin connects the junctions between endocardial cells and it is lost when EndMT occurs. Furthermore, VE-cadherin protein expressed in endocardial cells is a recognized endocardial cell marker. Niessen et al. (Circulation Research, 2008, doi: 10.1161/CIRCRESAHA.108.174318.) described a mechanism in which the function of Slug and Snail is similar, both involving the negative regulation of VE-cadherin expression and the initiation of EndMT. In this context, we agree with the reviewer’s position that VE-cadherin is a key player in EndMT and represses EndMT.

5) For Fig 5A it appears that the authors selected differnet p-AKT and AKT lanes for their analysis of wild-type, heterozygous and null embryos. The GAPDH control lanes were also selected form alternative lanes as well (see Supplemental Material). This is unacceptable.

Response: We agree with your opinion. We use Western blot data from the same lane in the revised Fig. 5A. 

6) Fig. 5B. Show the E10.5 immunohistochemistry.

Response: Fig. 5B and C only show the E11.5 immunohistochemistry and there are no images of E10.5 immunohistochemical staining. We performed immunofluorescence staining in both E10.5 and E11.5 embryos to measure the relative intensities of P-AKT and P-ERK proteins, but immunohistochemical staining was performed and imaged only at the representative E11.5 stage.

7) Fig. 5C. Show the E10.5 immunohistochemistry.

Response: Although immunofluorescence staining was performed in E10.5 and E11.5 embryos to measure P-AKT and P-ERK protein intensity, immunohistochemical staining was performed and imaged only at the representative E11.5 stage.

8) I agree that the authors show that, “In both mutants, the right ventricle was smaller than the left ventricle, and the myocardial thickness was reduced.” I however, disagree that these results, “suggest that the heart defects in both the Erbb3-/- and Erbb37A/7A mutants were caused by overall growth retardation.” If that were true, then you would expect an overall reduction in heart size, not in defects in only one ventricle. They may not have a good explanation, but no explanation is better than one that can’t be supported by the data.

Response: In our results, the right ventricle of both mutants was smaller than the left ventricle and had decreased myocardial thickness. Our results showing defects in cell proliferation and EndMT support this finding. In addition, although not quantified, Figure 2 shows that the heart size from E12.5 to E13.5 is progressively smaller than normal. These findings were the basis why we stated that heart defects in both Erbb3-/- and Erbb37A/7A mutants were caused by the overall growth retardation.

9) I also note that the change in the right ventricle size compared to the left ventricle is easy to see at E12.5, The authors show no embryo photos at that stage. Instead, the small embryo size is seen at E13.5 when the hearts in section also seem to be overall smaller than wild-type.

Response: Thank you for the query. In E12.5 embryos, there were no morphological differences between the normal and the two variants. Therefore, the heart sections of E12.5 and E13.5 shown in Figure 2 address the reviewer's concern.

10) Was the myocardial thickness of the left ventricle affected?

Response: We found decreased myocardial thickness in both Erbb3-/- and Erbb37A/7A, but the difference between the left and right myocardial thickness was not significant.

11) The authors state, “Overall, our results suggest that the YXXA mutation completely blocks the activation of ERBB3-dependent PI3K signaling than does the FXXM mutation of ERBB3.” This is not grammatically correct. Did they mean to say, “

 PI3K signaling more so than does the FXXM mutation of ERBB3.”

Response: The constructive feedback is appreciated. We have modified the phrase on p.14, lines 351–353, as follows: “Overall, our results suggest that the YXXA mutation blocks the activation of ERBB3-dependent PI3K signaling more than the FXXM mutation of ERBB3.”

12) The authors also state, “Furthermore, our data suggest that ERBB3-dependent PI3K signaling is essential for AV cushion development and valve formation.” The AV cushions develop in ERBB3 null mice, right? Hence, ERRB3 can’t be essential for AV cushion development. In addition, I have seen no evidence of any effect on valve formation in this paper.

Response: Thank you for raising this important point. We agree and have removed the phrase on p.14, lines 353–354.

---

## [Editor Report · Decision Letter 2]

20 Oct 2021

ERBB3-dependent AKT and ERK pathways are essential for atrioventricular cushion development in mouse embryos

PONE-D-21-03408R2

Dear Dr. Kim,

We’re pleased to inform you that your manuscript has been judged scientifically suitable for publication and will be formally accepted for publication once it meets all outstanding technical requirements.

Kind regards,

Robert W Dettman, PhD

Academic Editor

PLOS ONE

Additional Editor Comments (optional):

Thank you for your revised submission. I have reviewed your changes and found them to be acceptable. Thank you for your excellent submission.
---

## [Editor Report · Acceptance letter]

22 Oct 2021

PONE-D-21-03408R2 

ERBB3-dependent AKT and ERK pathways are essential for atrioventricular cushion
development in mouse embryos 

Dear Dr. Kim:

I'm pleased to inform you that your manuscript has been deemed suitable for publication in PLOS ONE. Congratulations! Your manuscript is now with our production department. 

Kind regards, 

on behalf of

Dr Robert W Dettman 

Academic Editor

PLOS ONE